# Helical structure motifs made searchable for functional peptide design

Cheng-Yu Tsai [1,2,3,18], Emmanuel Oluwatobi Salawu [1,4,5,18], Hongchun Li[1,6,7,8,18], Guan-Yu Lin [9,18], Ting-Yu Kuo[9,18], Liyin Voon[1,18], Adarsh Sharma [1], Kai-Di Hu[1], Yi-Yun Cheng[10], Sobha Sahoo [1], Lutimba Stuart[1], Chih-Wei Chen[1], Yuan-Yu Chang[1,10], Yu-Lin Lu [1], Simai Ke [1], Christopher Llynard D. Ortiz [1,11,12], Bai-Shan Fang[7,13], Chen-Chi Wu [3,14], Chung-Yu Lan [9,15✉], Hua-Wen Fu [9,15✉] & Lee-Wei Yang [1,4,16,17✉]

The systematic design of functional peptides has technological and therapeutic applications. However, there is a need for pattern-based search engines that help locate desired functional motifs in primary sequences regardless of their evolutionary conservation. Existing databases such as The Protein Secondary Structure database (PSS) no longer serves the community, while the Dictionary of Protein Secondary Structure (DSSP) annotates the secondary structures when tertiary structures of proteins are provided. Here, we extract 1.7 million helices from the PDB and compile them into a database (Therapeutic Peptide Design database; TP-DB) that allows queries of compounded patterns to facilitate the identification of sequence motifs of helical structures. We show how TP-DB helps us identify a known purification-tag-specific antibody that can be repurposed into a diagnostic kit for *Helicobacter pylori*. We also show how the database can be used to design a new antimicrobial peptide that shows better *Candida albicans* clearance and lower hemolysis than its template homologs. Finally, we demonstrate how TP-DB can suggest point mutations in helical peptide blockers to prevent a targeted tumorigenic protein-protein interaction. TP-DB is made available at http://dyn.life.nthu.edu.tw/design/.

[1] Institute of Bioinformatics and Structural Biology, National Tsing Hua University, Hsinchu 300044, Taiwan. [2] Graduate Institute of Medical Genomics and Proteomics, National Taiwan University College of Medicine, Taipei 100025, Taiwan. [3] Department of Otolaryngology, National Taiwan University Hospital, Taipei 100225, Taiwan. [4] Bioinformatics Program, Institute of Information Sciences, Academia Sinica, Taipei 115201, Taiwan. [5] Machine Learning Solutions Lab, Amazon Web Services (AWS), Herndon, VA, USA. [6] Research Center for Computer-Aided Drug Discovery, Shenzhen Institutes of Advanced Technology, Chinese Academy of Sciences, 518055 Shenzhen, China. [7] College of Chemistry and Chemical Engineering, Xiamen University, 361005 Xiamen, China. [8] Department of Computational and Systems Biology, School of Medicine, University of Pittsburgh, Pittsburgh, PA 15213, USA. [9] Institute of Molecular and Cellular Biology, National Tsing Hua University, Hsinchu 300044, Taiwan. [10] Praexisio Taiwan Inc., New Taipei 221425, Taiwan. [11] Chemical Biology and Molecular Biophysics Program, Institute of Biological Chemistry, Academia Sinica, Taipei 115201, Taiwan. [12] Department of Chemistry, National Tsing Hua University, Hsinchu 300044, Taiwan. [13] The Key Laboratory for Chemical Biology of Fujian Province, Key Lab for Synthetic Biotechnology of Xiamen City, Xiamen University, 361005 Xiamen, China. [14] Department of Medical Research, National Taiwan University Hospital Hsin-Chu Branch, Hsinchu 302058, Taiwan. [15] Department of Life Science, National Tsing Hua University, Hsinchu 300044, Taiwan. [16] Physics Division, National Center for Theoretical Sciences, Taipei 106319, Taiwan. [17] PhD Program in Biomedical Artificial Intelligence, National Tsing Hua University, Hsinchu 300044, Taiwan. [18] These authors contributed equally: Cheng-Yu Tsai, Emmanuel Oluwatobi Salawu, Hongchun Li, Guan-Yu Lin, Ting-Yu Kuo, Liyin Voon. ✉email: cylan@life.nthu.edu.tw; hwfu@life.nthu.edu.tw; lwyang@life.nthu.edu.tw

Secondary structures comprise a subset of amino acid sequences where each constituent residue forms hydrogen bonds (H-bonds) in specific orders with other constituent residues, at their peptidyl backbones, which grants a special mechanical property in our life-thriving nanomachines—proteins. Particularly, α-helices, where the $i$-th residue in the helix H-bonded with $i + 4$-th residue, mediate various types of molecular functions including cell membrane penetration[1–4], subcellular localization[5], intramolecular/intermolecular allostery[6–9], special helix-DNA scaffolds[10], and protein–protein interaction[7,11], provided that the primary sequence meets both the structural (main chain and side chain) and chemical (side chain) requirements for specific functions of interest. Especially at the interface, either water-lipid interface at the cell membrane, or protein-DNA/protein–protein interface, these helices have disparate physicochemical properties at their two sides to properly play their functional roles. Take CM15[12,13], an amphiphilic, helical antimicrobial peptide (AMPs) for example, for its proper bactericidal function, the positively charged lysines and hydrophobic residues need to stay at opposite sides of a helical wheel, the 2D projection of amino acids that one would see if viewed from a helix's longitudinal axis. As a result, after Lys3 (K3), there would not be any positively charged residues till K6 and K7 appear at the same side (Fig. 1 in the publication of Sato and Feix[12]), or after Leu4 (L4) and Phe5 (F5), there are no hydrophobic residues till Ile8 (I8) is present at the same side. In this example, what the peptide needs for proper function is a stretch of amino acid sequence consisting of a pattern of (K3)$x_1x_2$(K6) or (F5)$x_3x_4$(I8), where $x_1$ and $x_2$ ($x_3$ and $x_4$) can be any hydrophobic (positively charged) amino acid that still holds a helical structure. From the point of view of protein design, either to design a better AMP[2] or to propose stronger interface blockers[11] for therapeutic purposes, it is not an easy task to allow constituent residues to address physicochemical needs, meeting a specific sequence pattern, while maintaining the required structural integrity. Suitable tools need to be developed to facilitate the process.

Currently, both (A) pattern-based (only) search engines locating desired functional motifs in primary sequences regardless of their evolutionary conservation and (B) dedicated secondary structural databases indexed for (A)'s purpose are lacking. Protein Secondary Structure database (PSS) proposed 30 years ago no longer serves the community[14], while the Dictionary of Protein Secondary Structure (DSSP)[15] annotates the secondary structures when tertiary structures of proteins are provided[16]. However, both DSSP and protein data bank (PDB) do not provide a search engine for specific types of secondary structures (say, the entire query sequence should belong to an alpha-helix) for >170,000 structurally solved sequences, nor do they provide any search function that recognizes specific sequence patterns (see the beginning of "Methods" for our definition of "patterns"). To better integrate functional structural fragments into therapeutics[17–19], we extracted ~1.7 million (1,676,117) structurally resolved helices and their contacting/interacting helical partners from the PDB, from which helical propensity (HP) and coordination (contact) number are derived (Table 1). We then establish a pattern-specific search engine (see below) to find indexed sequences bearing particular spatio-chemical properties instead of evolutionary homology (say, the purpose of Pattern Hit Initiated BLAST (PHI-BLAST)[20] that takes the input of both residue pattern and the template sequence). We use this Therapeutic Peptide Design dataBase (TP-DB) to develop a new anti-fungal peptide/AMP, helix–helix interface blockers that could potentially disrupt the interaction between Shugoshin-1 (Sgo1) and protein phosphatase 2A (PP2A) and a diagnostic reporter to detect *Helicobacter pylori* infection.

**Table 1 Derived helical propensity for each amino acid in descending order.**

| AA | $HP_{a.a.}$ | $NA_{a.a.}$ | $HP_{NA}$ ($HP_{a.a.}/NA_{a.a.}$) | Helical propensity, log ($HP_{NA}$) |
|---|---|---|---|---|
| Alanine (A) | 0.109 | 0.083 | 1.320 | 0.277 |
| Tryptophan (W) | 0.014 | 0.011 | 1.284 | 0.250 |
| Glutamate (E) | 0.084 | 0.067 | 1.246 | 0.220 |
| Leucine (L) | 0.115 | 0.097 | 1.192 | 0.175 |
| Glutamine (Q) | 0.045 | 0.039 | 1.145 | 0.135 |
| Tyrosine (Y) | 0.033 | 0.029 | 1.130 | 0.122 |
| Lysine (K) | 0.064 | 0.058 | 1.100 | 0.095 |
| Methionine (M) | 0.026 | 0.024 | 1.079 | 0.076 |
| Arginine (R) | 0.059 | 0.055 | 1.067 | 0.065 |
| Phenylalanine (F) | 0.038 | 0.039 | 0.984 | −0.016 |
| Isoleucine (I) | 0.058 | 0.059 | 0.978 | −0.022 |
| Histidine (H) | 0.022 | 0.023 | 0.969 | −0.031 |
| Aspartate (D) | 0.052 | 0.055 | 0.952 | −0.049 |
| Asparagine (N) | 0.038 | 0.041 | 0.936 | −0.066 |
| Threonine (T) | 0.049 | 0.053 | 0.918 | −0.086 |
| Valine (V) | 0.062 | 0.069 | 0.902 | −0.103 |
| Cysteine (C) | 0.012 | 0.014 | 0.876 | −0.132 |
| Serine (S) | 0.053 | 0.066 | 0.804 | −0.218 |
| Glycine (G) | 0.046 | 0.071 | 0.650 | −0.431 |
| Proline (P) | 0.021 | 0.047 | 0.445 | −0.810 |

*AA* amino acid, *$HP_{a.a.}$* helical propensity for each amino acid, *$NA_{a.a.}$* natural abundance of each amino acid, *$HP_{NA}$* $HP_{a.a.}$ normalized by $NA_{a.a.}$.

## Results

**Database features and statistics**. TP currently comprises 1,676,117 helical peptides (24,301,682 amino acids in total) extracted from 130,000+ experimentally solved protein structures stored in PDB. The apparent goal of establishing search engines for protein secondary structures (starting from all types of helical configurations, e.g., α-helices and $3_{10}$ helices) is to find stretches of protein sequences that match the queried pattern and adapt a helical conformation. Since protein sequences assume structures in the context of their 3-dimensional environment full of residue contacts, it is quite likely the helices are not as helical as when they are part of their parent proteins. We herein design two metrics to evaluate and prioritize the peptides, in the query results, which are highly likely to still stay helical when expressed or synthesized alone—HP and contact number. A higher HP and/or lower contact number would suggest a good helicity for peptides even in their isolated forms. For the former, we use a normalized HP (see "Methods"), $HP_{NA}$, defined as $HP_{a.a.}/NA_{a.a.}$, where $HP_{a.a.}$ is the HP of a given amino acid type and $NA_{a.a.}$ is natural abundance of that amino acid (Table 1). Both $HP_{a.a.}$ and $NA_{a.a.}$ are probabilities, where $HP_{a.a.}$ is defined as the fraction of a certain amino acid type in TP-DB and $NA_{a.a.}$ is the fraction of the same type of amino acid (https://web.expasy.org/protscale/pscale/A.A.Swiss-Prot.html) from the ExPASy tool "ProtScale"[21]. A log-odds ratio, log ($HP_{NA}$) or log ($HP_{a.a.}/NA_{a.a.}$) is found for each amino acid type, and the reported HP for each peptide in TP-DB is the sum of log ($HP_{NA}$) for every constituent residue in this peptide (Table 2). The second metric is the average contact number, defined as the total number of amino acids, represented by their $C_\alpha$ atoms, contacting a helical peptide of interest in TP-DB within 7.3 Å[22], divided by the length of the peptide. The lower contact number of a helix in its parent protein suggests a higher probability for it to stay in the helical form in isolation. This assumption lies on that a highly contacted helix could be formed only when it is stabilized by such a high contact, supported by its coordinating residues in spatial proximity, although the helix could still be helical in isolation (see below). The distributions of

**Table 2 Results obtained from the TP-DB for the query [W 2 W 2 W].**

| # | U# | Matched Sequence (MS) | Matched Pattern | Full Helix (FH) | PDB ID: Chain | Positions in PDB (MS),(FH) & File Download | Helical Propensity↕ | Contact↕ | Interacting Partners | Helicity%↕ |
|---|----|----|----|----|----|----|----|----|----|----|
| 1 | 1 | WLEWIRW | W2W2W | DVQQLLIWLEWIRWESD | 4eba:F | (287,293) (280,296) | 1.189 | 8.857 | 4eba:F (324, 338) | 0.360 |
| 10 | 2 | WQQWLNW | W2W2W | DLATFWQQWLNW | 3tsu:A | (651,657) (646,657) | 1.131 | 7.857 | | 0.392 |
| 11 | 3 | WQRWENW | W2W2W | VNSYLWQRWENWFWNVTLR | 2lzq:A | (10,16) (5,23) | 1.105 | 8.000 | | 0.385 |
| 12 | 4 | WDAWLNW | W2W2W | WDAWLNWFR | 1z8h:D | (198,204) (198,206) | 1.089 | 7.286 | | 0.410 |
| 13 | 5 | WKCWARW | W2W2W | NVEDWKCWARWRLIRARA | 3zuk:B | (283,289) (279,296) | 1.055 | 9.143 | 3zuk:B (415, 434) | 0.339 |
| 15 | 6 | WAWWISW | W2W2W | AGEWLGSWTIFYWAWWISWSPFVGMFLAR | 4llh:A | (371,377) (359,387) | 1.039 | 10.571 | 4llh:A (449, 479) 4llh:A (511, 525) 4llh:A (527, 545) | 0.285 |
| 25 | 7 | WVDWWQW | W2W2W | QWVDWWQWWVK | 4n7k:L | (259,265) (258,268) | 0.985 | 7.286 | 4n7k:M (81, 88) 4n7k:M (178, 192) | 0.402 |
| 86 | 8 | WKDWESW | W2W2W | SVLRKALHDSLHDCSHWFYTRWKDWESWYSQSF | 3zrh:A | (481,487) (460,492) | 0.799 | 8.857 | 3zrh:A (522, 533) | 0.330 |
| 87 | 9 | WTAWSTW | W2W2W | WTAWSTWKYC | 3cb7:A | (106,112) (106,115) | 0.638 | 7.714 | | 0.360 |
| 88 | 10 | WPEWWNW | W2W2W | GWPEWWNWWLE | 3wmm:L | (268,274) (267,277) | 0.345 | 8.857 | 3wmm:M (82, 90) 3wmm:M (180, 193) | 0.296 |
| 89 | 11 | WPEWWGW | W2W2W | WPEWWGWWL | 7prc:L | (259,265) (259,267) | -0.020 | 8.429 | | 0.284 |

As shown in https://dyn.life.nthu.edu.tw/design/result?JobID=6186089an, a total of 105 helical peptides can be found with the query [W 2 W 2 W] and their "W\*\*W\*\*W" motifs are shown in blue with 3 equally spaced tryptophans marked in bold. Only the unique 11 sequences (indexed in the U# column) are shown here and the red block highlights the new helical stretch we used for the AMP design.

the HP, log ($HP_{NA}$), and the average contact number of all the TP-DB helices can be found in Supplementary Fig. 2, with an average ± standard deviation of $0.38 \pm 0.85$ and $8.87 \pm 0.96$, respectively. The two quantities are independent, with a negligible correlation coefficient of $-0.08$ (Supplementary Fig. 2c).

**Construction of the search engines for TP-DB.** A pattern-based search engine without considering sequence homology is subsequently established and tailored to the TP-DB's data structure. Helical peptides that match specific patterns are first indexed in the database. A pattern could be of the form "Y\*\*\*G\*\*K", where "\*" belongs to any of the 20 amino acids and Tyrosine (Y), Glycine (G), and Lysine (K) are called 'anchoring residues' of the pattern. This is equivalent to the regular expression "Y-x(3)-G-x(2)-K" in the PROSITE format and also used in the PHI-BLAST[20] search. The location of a pattern is indexed by the PDB ID, the chain ID, and the position where the pattern begins in that chain (see below). Considering both flexibility and search comprehension, we let the constituent pattern be [$A_1$ m $A_2$ n $A_3$] where $A_1$, $A_2$, and $A_3$ are the single-letter codes for amino acids, in between which are the spacings represented by the whole numbers m and n. We currently allow m and n to be from zero to four. A key "Y3G2K" or its search pattern [Y 3 G 2 K] would indicate a "Y" is $(3 + 1)$ residues upstream to a "G" that is $(2 + 1)$ upstream to a "K". A key is paired with identifiers (values) referring to where this key can be found in all the stored peptides (Fig. 1c, e). Specifically, these values, stored in an array, are peptide identifiers and all the starting positions of a certain key in each of the peptides (Fig. 1c). These key–value pairs comprise our database (Fig. 1), TP-DB (http://dyn.life.nthu.edu.tw/design/),

which could be searched by a simple pattern [$A_1$ m $A_2$ n $A_3$] or a pattern compounded by several of these simple patterns (Fig. 1f).

To query the database, users need to specify at least three anchor residues and the gaps that separate them, such that "ADE" or "K\*\*\*\*VA" (Alanine: A; Aspartate: D; Glutamate: E; Valine: V) can be queried by [A 0 D 0 E] and [K 4 V 0A], respectively (Fig. 1d). For more complicated queries, say "A/Y 3 G 2 K 3 H 4 K" (Histidine: H), it is first broken down into a combination of two subqueries "A 3 G 2 K 3 H 4 K" and "Y 3 G 2 K 3 H 4 K" (Fig. 1f). The five anchor residues in each subquery are treated as two concatenated three-anchor keys. The matched patterns co-localized in a peptide sequence are further parsed to take into account the overlapped regions before returning the results. TP-DB currently has two search engines to search the indexed peptides, one by pattern search, described above, and the other by BLAST (parameterized for short sequences). The use of the latter is beyond the scope of this paper.

**Anti-FLAG antibody is repurposed for pathogen diagnosis.** To demonstrate our search engine's utility of finding patterns without considering residues' evolutionary similarity, we would like to identify the sequences matching the queried pattern which can only be found by our TP-DB but not by other tools, including PHI-BLAST that considers both the pattern matching and evolutionary similarity between residues in the sequence alignment. The latter has been the main (if not only) bioinformatics tool to find biological sequences that match a specific pattern while inferring homology[23]. To showcase the prowess and disparate utility of the TP-DB, we demonstrate below how a purification tag (herein FLAG-tag) and its antibody (herein M2

**a**

Peptide _P_: ADEKKFWGKYLYEVA          Peptide _z_: TAFEGGILKKGHHCSYTKH
Positions:  012345678901234          Positions:  0123456789012345678

**b**

```
Keys in Peptide P                    |    Start Positions in Peptide P
A0D0E, D0E0K, ..., Y0E0V, E0V0A      |    0, 1, ..., 11, 12
A0D1K, D0E1K, ..., L0Y1V, Y0E1A      |    0, 1, ..., 10, 11
...                                  |    ...
A0D4W, D0E4G, ..., G0K4V, K0Y4A      |    0, 1, ..., 7, 8
...                                  |    ...
A4F0W, D4W0G, ..., G4E0V, K4V0A      |    0, 1, ..., 7, 8
...                                  |    ...
A4F4L, D4W4Y, ..., K4K4V, K4Y4A      |    0, 1, ..., 3, 4
```

**c**

```
DBINDEX = { A0D0E: [ [PeptideP , 0], [PeptideP-1 , 17, 21, ...], ...],
            D0E0K: [ [PeptideP , 1], [PeptideP-9 , 13, 25, ...], ...],
            ...
            K4V0A: [ [PeptideP , 8], [PeptideP-78 , 1, 64, ...], ...],
            ...
            K4Y4A: [ [PeptideP , 4], [PeptideP-2 , 23, 41, ...], ...],
            ... }
```

**d**

```
Sample patterns      Keys in        Examples of peptides that
of interest          the DB         the patterns can match

ADE                  A0D0E          ADEKKFWGKYLYEVA

DEK                  D0E0K          ADEKKFWGKYLYEVA

K----VA              K4V0A          ADEKKFWGKYLYEVA

K----Y----A          K4Y4A          ADEKKFWGKYLYEVA
```

**e**

```
Simple Query: K 4 Y 4 A, which is equivalent to K4Y4A or K----Y----A
Direct Fetch: K4Y4A from DBINDEX in the RAM: →
              [ [PeptideP , 4], [PeptideP-2 , 23, 41, ...], ...],
```

**f**

[1] Complex Query: Process into a Combination of Simple Queries

| Example of Complex Queries | Matching Patterns | Constituent Simple Queries |
|---|---|---|
| A 3 G 2 K 3 H 4 K | A 3 G 2 K 3 H 4 K | A 3 G 2 K<br> K 3 H 4 K |
| A/Y 3 G 2 K 3 H 4 K | A 3 G 2 K 3 H 4 K | A 3 G 2 K<br> K 3 H 4 K |
|  | Y 3 G 2 K 3 H 4 K | Y 3 G 2 K<br> K 3 H 4 K |
| A 2,3 G 2 K 3,4 H 4 K | A 2 G 2 K 3 H 4 K | A 2 G 2 K<br> K 3 H 4 K |
|  | A 2 G 2 K 4 H 4 K | A 2 G 2 K<br> K 4 H 4 K |
|  | A 3 G 2 K 3 H 4 K | A 3 G 2 K<br> K 3 H 4 K |
|  | A 3 G 2 K 4 H 4 K | A 3 G 2 K<br> K 4 H 4 K |

[2] Simple Queries' results are aggregated into the required results

antibody) can be repurposed into a diagnostic kit for detecting human pathogens. The relevant experimental methods are listed in Supplementary Information. FLAG-tag is known to have a sequence DYKDDDDK[24] with the pattern D-Y-K-x-x-[DE] being experimentally confirmed as the main affinity determinant motif[25]. "DYK" here is the strongest determinant while the last D (or E) is of secondary importance. We would like to search

TP-DB for proteins that (i) are not included in the PHI-BLAST search results and (ii) belong to proteins in human pathogens but not in non-pathogenic bacteria.

When querying the pattern "D 0 Y 0 K 2 D/E" against the TP-DB, we could identify totally 93 sequences with unique 19 sequences. While using PHI-BLAST to search NCBI's non-redundant protein sequences (nr) with the pattern D-Y-K-x(2)-[DE] (in PROSITE

**Fig. 1 Indexing the peptides and querying of TP-DB. a** The positions of amino acids in the $p$-th helical peptide (peptide$_p$) are indexed from 0 to 14. **b** All the possible patterns that can be found in peptide$_p$, stored as keys and values. **c** In TP-DB (or DB$_{Index}$), keys are paired values, and values are where these keys can be found, including in which peptides and the starting position of the keys. For instance, [A 0 D 0 E] can be found in the "0-th position" in the $p$-th peptide, while it can also be found in the 17th and 21st positions in the $(p+1)$-th peptide. **d** The patterns of interest are translated into "keys" comprising amino acids' single-letter codes separated by numbers, the sizes of the gaps. **e** Results for a simple query retrieved from TP-DB. **f** A compounded query can be broken down into subqueries and the result is a joint result from subqueries. For efficient search, two types of libraries are loaded in a timely order. What is constantly loaded in the memory, with a small memory footprint, is a collection of key–value pairs that only index which proteins have certain keys but not their locations. Only when certain proteins are visited because they contained the queried keys, then libraries of a second type, inferring the locations of keys in certain proteins, are then loaded into the memory to report the found sequences, before the protein-relevant libraries are unloaded from the memory again soon after a search job is finished. All abbreviations of amino acids are listed in Table 1.

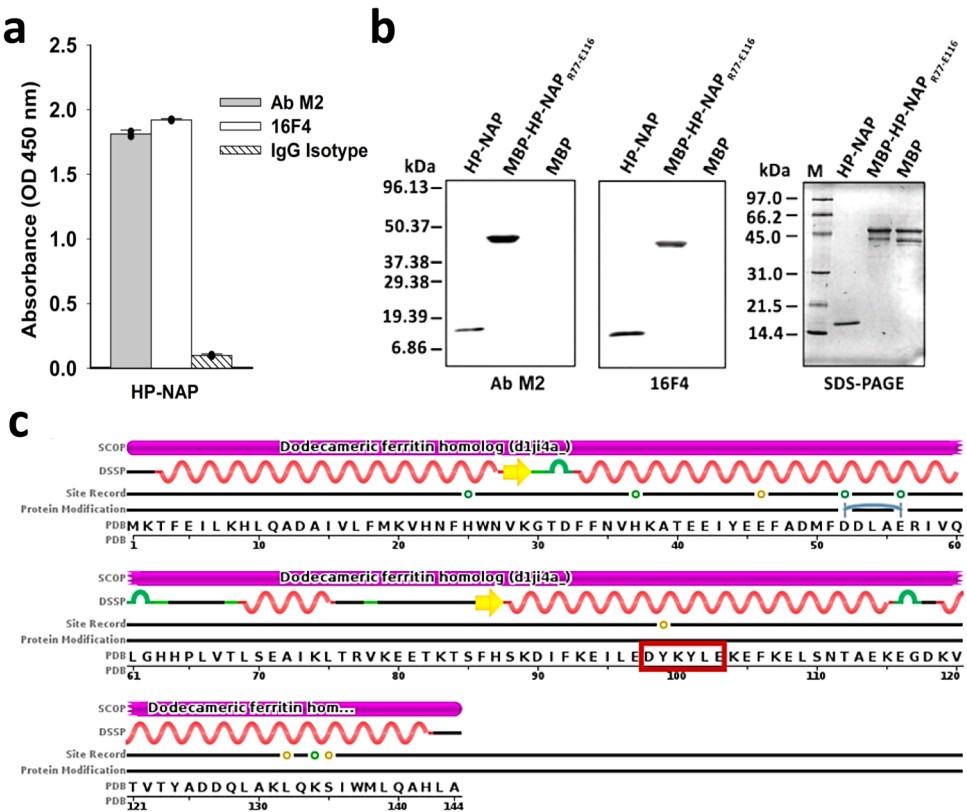

**Fig. 2 Detection of HP-NAP by anti-FLAG M2 antibody. a** Detection of HP-NAP by anti-FLAG M2 antibody using recombinant HP-NAP-based ELISA. Recombinant HP-NAP was purified by a small-scale DEAE Sephadex negative mode batch chromatography. Purified HP-NAP as an antigen was coated on an ELISA plate as described in "Methods." The anti-FLAG M2 antibody, its corresponding mouse IgG antibody, and the antibody 16F4 against HP-NAP, as a positive control, were subjected to recombinant HP-NAP-based ELISA. The result is expressed as absorbance at OD$_{450\,nm}$. Data are presented as mean ± standard deviation (S.D.) of one experiment in duplicate (black dots are individual results). Similar results were obtained in two independent experiments. The difference between IgG isotype and any of the other two groups are statistically significant ($p = 0.00017$ between M2 and IgG; $p = 0.00004$ between 16F4 and IgG) according to Student's $t$ test (two-tailed), and the source data are provided as a Source data file. **b** Detection of recombinant HP-NAP by anti-FLAG M2 antibody using western blot analysis. Recombinant HP-NAP was purified by two consecutive gel-filtration chromatography. MBP-tagged HP-NAP$_{R77-E116}$ and MBP were partially purified by the MBPTrap HP column using the ÄKTA Purifier. These purified proteins, 1 µg each, were subjected to SDS-PAGE on a 15% gel and western blot analysis with anti-FLAG M2 antibody and the antibody 16F4. Molecular masses (M) in kDa are indicated in the left of the blots and the gel. Similar results were obtained in two independent experiments. **c** The sequence of HP-NAP. Note that "DYKYLE," highlighted in the red box, matches the pattern D-Y-K-x-x-[DE] and is included in the R77-E116 stretch. The image is downloaded from PDB website (code:1JI4) with minor modifications.

format) and FLAG sequence "DYKDDDDK", we were not able to find any sequence with an $E$-value <100. On the other hand, among the 19 TP-DB-identified sequences, HP-NAP, containing the sequence "DYKYLE", can be found in *H. pylori* strain 26695 (accession no. AE000543, ATCC) but not in regular non-pathogenic bacteria. If the anti-FLAG M2 antibody can recognize this randomly selected pathogenic protein for its containing a FLAG-determinant motif (herein D-Y-K-x-x-[DE]), we can potentially use the same concept to repurpose the original use (say, purification) of any

known pair of antibody and antigen recognition motif (say, M2 antibody and D-Y-K-x-x-[DE]) into its alternative use, herein the potential diagnostic kit for human pathogens. As expected, the anti-FLAG M2 antibody was indeed found to be capable of recognizing HP-NAP as analyzed by enzyme-linked immunosorbent assay (ELISA; Fig. 2a) and western blot (Fig. 2b). Further analysis showed that the M2 antibody detected maltose-binding protein (MBP)-tagged HP-NAP$_{R77-E116}$ but not MBP (Fig. 2b), indicating that M2 antibody binds to the polypeptide fragment containing the residues

Arg77 to Glu116 of HP-NAP. The antigen recognition motif, DYKYLE, is present in this polypeptide fragment as shown in Fig. 2c.

As a result, the antibody (herein M2 antibody) of a purification kit can be repurposed into a diagnostic kit for detecting human pathogens, herein *H. pylori*, through recognizing native HP-NAP by ELISA or the HP-NAP monomer as a 17 kDa band by western blot.

**α-Helical AMP design**. Pan-drug-resistant bacteria are no less a threat to mankind than viral infections that could cause a pandemic[26]. A discouraging forecast has suggested that by 2050, 10 million people could die from these superbugs each year, with half of them in Asia[27]. AMPs that target bacterial lipid membranes, evolving at a much slower pace than bacterial protein receptors, appear to be a promising alternative for antibiotics. There have been substantial efforts made to study AMPs' insertion mechanism[28–31] and analyze the determining characteristics for their efficacy[32,33]. To join these collective efforts and showcase the good use of TP-DB, we first examined in detail the insertion processes of two previously reported helical AMPs, W3_p1 (sequence: KK WRK WLK WLA KK)[34] and W3_p2 (sequence: KK WLK WLK WLK KK)[2] by MD simulations. The two amphiphilic AMPs, membrane-selective and mediated through no known protein receptors, have three equally spaced tryptophans at their hydrophobic face, while positively charged lysines/arginines are on the other side, which is a typical feature and requirement for helical AMPs. The goal of the simulations is to understand each residue's physicochemical role in the bacteria-membrane insertion process while earlier reports[2,34–37] have suggested the deeper a peptide penetrates the lipid membrane, the higher the bactericidal potency it can have.

**Equally spaced three tryptophan residues aligned on the same side of the helix are mechanistically essential to AMP insertion**. With microsecond simulations, we revealed types of contact modes and time-resolved atomic interaction between AMPs and lipid bilayers before AMPs' final insertion into the bacterial membrane (see Supplementary Movie 1 and Supplementary Fig. 3). Taking W3_p1 for example, as shown in Fig. 3, lysine and arginine with long positively charged side chains preferentially interact with the phosphate groups in PC and PG (see Supplementary Fig. 1). These interactions not only "raise" some interacting lipid molecules out of the upper leaflet but also push away these phosphate groups due to the long arms of these positively charged side chains. This creates what we call a "hydrophobic cradle," which is a temporary vacuum created due to the aforementioned Lys/Arg-lipid interaction where at the bedding of the cradle lie the exposed aliphatic lipid tails whose head groups have been pushed either sideway or upward. One can see such cradles form at 46, 673, 678, and 715 ns in Fig. 3. This provides an important opportunity for adjacent tryptophan residue to flip in to enjoy such a hydrophobic interaction only when the orientation is right. Either W3 (Trp-3) or W9 (Trp-9) (see 46 ns in Fig. 3) close to N- or C-terminal double lysine residues can be the first tryptophan to take advantage of these hydrophobic cradles for insertion. However, W9 insertion near the C-terminus is transient (starting right in the beginning, at 10 ns, drifting away at 46 ns, resuming the insertion at 93 ns but then drifting away again soon after) (Supplementary Movie 1 and Supplementary Fig. 3). It cannot be maintained because the horizontal insertion pops the other two tryptophan up or sideways toward the bulk solvent. On the other hand, W3-initiated insertion benefits not only from the transiently formed hydrophobic cradles but also from the downward-facing N-terminus (see 530 ns, 715 ns and thereafter

in Fig. 3) that further brings W6 to contact the deeper part of a lipid (e.g., 715 ns and thereafter). The N-terminus- rather than C-terminus-initiated penetration is consistent with the aforementioned experimental results[38] and also common to most of the membrane anchoring signals[39]. N-terminus seems to better interact with the phosphate groups of lipids (e.g., at 678 ns) and transiently interact with the lipid tails better than the C-terminus.

As W3 starts to insert, its irreplaceable chemical importance in membrane insertion becomes immediately apparent in these time-resolved snapshots. As shown at 678 ns in Fig. 3, it can first interact with the choline groups of lipids through the cation–π interaction. Then, it can interact with the phosphate groups (715 ns) or the deeper side of the glycerol moiety (735 ns) in the same or different lipids via hydrogen bonds. Eventually, W3 reaches and interacts stably with lipid tails due to the hydrophobic interaction (741 ns onward). As W3 reaches the lipid phosphate groups or glycerol moieties, W6 (Trp-6) can start to interact with the choline group of the same lipid (e.g., 715 ns). When W6 starts to reach the phosphate group (735 ns) and lipid tails (741 ns), the originally vertical W3_p1 starts to lean down with the N-terminus still diving a bit deeper than the C-terminus, which brings W9 closer to the lipid on the sideway. At 756 ns, we see that W9 starts to interact with two choline groups of lipids. As W9 starts to sink deeper to interact with the phosphate group of a lipid (e.g., at 866 ns), to allow W3 and W6 being still stably anchored in the lipid tails, the N-terminus starts to open up and becomes less helical (866 ns onward), which helps split W3 and W6 further apart and bring W9 to rotate further downward and better align W6 and W9 on the same side. This also synchronizes with positively charged residues being upward-facing and sticking their side chains toward the bulk solvent (968 ns onward).

In summary, W3_p1 first interacts with lipids via electrostatic interaction with the phosphate groups of the lipids using the positively charged side chains of lysines, which exposes tryptophans, aligned in one side of the helix, to the bulk solvent. As N-terminus dives downward in the hydrophobic cradle, created by the long positively charged side chains to push the lipid heads away and expose their lipid tails, the W3 is flipped and interacts straightly with the deeper part of the lipids, which makes the AMP stand vertically on the lipid surface. As W6 further crawls into the opening and interacts with phosphate groups of the lipids, or deeper, the AMP starts to lean down with N-terminus, in turn, to become "loopy" and less helical, which helps W9 be better aligned with W6 and insert deeper. As also evident in the Supplementary Movie 1, within the 1 microsecond, all the 3 tryptophan residues in W3_p1 are inserted into the bacterial membrane while W3_p2, which is strictly helical, has only two tryptophan residues (W3 and W6) inserted into the membrane (especially after 800 ns). In short, these AMPs insert the bacteria membrane heavily relying on the three tryptophan anchors equally spaced with two amino acids.

**TP-DB suggests a helical motif physicochemically suitable for the design of a potent AMP with low cytotoxicity**. Our AMP design starts from known helical AFPs/AMPs serving as design templates. Based on these existing templates and key anchoring residues, herein the three equally spaced tryptophans (W3, W6, and W9), we replace residues flanked by these anchoring residues using TP-DB to find functionally favored substituents that are ensured to have desired secondary structure. The pattern "W**W**W" or [W 2 W 2 W] as the input query in TP-DB is used as the search query. We are interested in finding other helical peptides in TP-DB with the following three characteristics—(1) containing the W2W2W motif, (2) maintaining helicity in

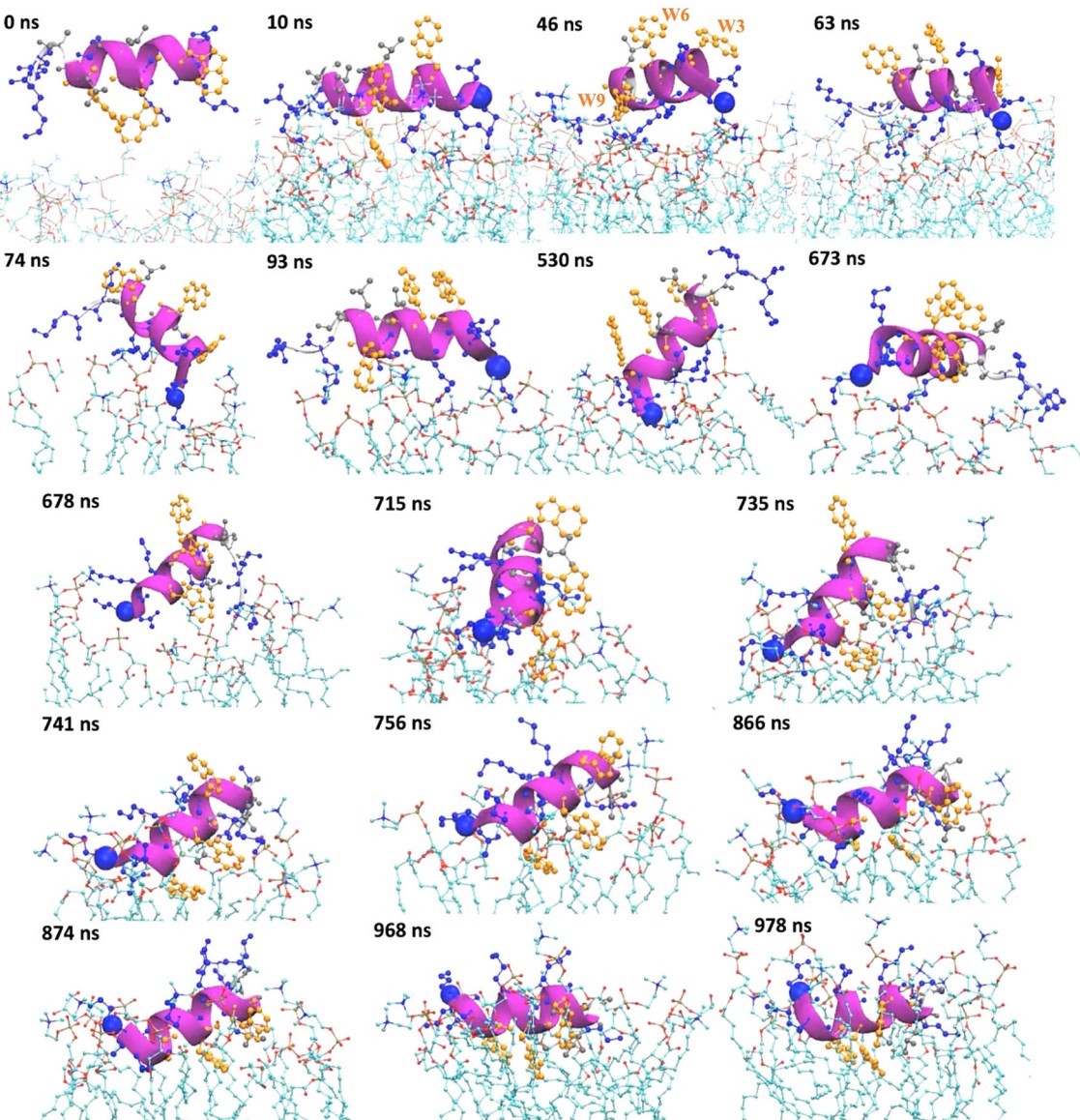

**Fig. 3 Mechanistic details of W3_p1's insertion into the bacterial membrane revealed by 1 μs MD simulations.** The Cα atom of the first residues (N-terminus) is highlighted with a blue VDW ball, from which the three tryptophan residues, shown in golden ball and stick, are W3, W6, and W9 (see also the panel of 46 ns). The positively charged residues in the AMP are represented in blue CPK. Starting from the 74th nanosecond, only the lipids staying within 3 Å from the AMPs are shown while part of their aliphatic tails can be removed for clarity. The heavy atoms of the lipids are represented in thin CPK; O, P, and C atoms are colored in red, brown, and cyan, respectively. The N atoms in the choline groups of lipids that engage cation–π interactions with the tryptophan are colored in blue.

isolation (with relatively high HP), and (3) enriched with positively charged residues so that the corresponding part in W3_p1 or W3_p2 can be replaced by the TP-found helical stretches (Table 2) to preserve the terminal lysine anchors.

One hundred and five helical peptides from different PDB IDs (and chain IDs) containing 11 unique "W**W**W" motifs can be found through the query [W 2 W 2 W] (Table 2). The peptide "WKCWARWRL" (the 5th in the #U column; Cysteine: C) having the most positively charged residues and a relatively high HP is chosen as our top AMP candidate. The peptide is derived from the mycobacterium tuberculosis Zinc-dependent metallo-protease-1 (Zmp1) (PDB: 3ZUK), which is a soluble protein and does not associate with the membrane. The peptide sits at the relatively exterior side of the protein but is not wide open to the bulk solvent. The flanking double Lys residues are further added to extend this motif into a full-fledged AMP candidate. This

newly designed AMP, named W3_db5, is then investigated by MD simulations and synthesized for the subsequent bactericidal experiments.

Contrasting to the simulation results of W3_p1 and W3_p2, W3_db5 has only one Trp (W3) inserted into the bacterial membrane by the end of 1 μs simulation (Supplementary Movie 1 and Supplementary Fig. 3). To further examine the bactericidal and anti-fungal activities of these AMPs, we measure the minimal inhibitory concentration ($MIC_{90}$) at which the growth of 90% of the pathogenic fungus *Candida albicans* SC5314 and *Escherichia coli* ATCC 25922 can be inhibited. Experiments are performed in pentaplicate and $MIC_{90}$ is determined as the majority results of the 5 replicates (see "Methods"). It is worth mentioning that we could reproduce the reported $MIC_{90}$ values for three published AMPs when using the same culture protocol to culture *E. coli* ATCC 25922. They are CM15, Anoplin-R5KT8W, and FK13 with

**Table 3 The anti-fungal, anti-bacterial activity, and hemolytic toxicity of TP-DB-designed AMP (W3_db5) and other AMPs containing the "W2W2W" motif.**

| Name | Sequence | MIC$_{90}$ vs. *C. albicans* (μg/ml) | MIC$_{90}$ vs. *E. coli* (μg/ml) | MHC$_5$ (μg/ml) |
|---|---|---|---|---|
| W3_p1 | KK **W**RK **W**LK **W**LA KK | 7.5 | 15 | 120 |
| W3_p2 | KK **W**LK **W**LK **W**LK KK | 7.5 | 15 | 90 |
| W3_db5 | KK **W**KC **W**AR **W**RL KK | 3.75 | 30 | >>240 |
| W3_n1 | KK **W**GN **W**GG **W**RL KK | N/A | >>120 | N/A |
| W3_n2 | KK **W**KD **W**ES **W**RL KK | N/A | >>120 | N/A |

MIC$_{90}$ means the minimal concentration of AMP to inhibit 90% growth of pathogens, while MHC$_5$ means the minimal concentration of AMP to induce 5% hemolysis against human erythrocytes. The three equally spaced tryptophans (W3, W6, and W9) are marked in bold and underlined.

reported MIC values of 4.0[13], 15.1[40], and 27.5[41] μg/ml, respectively, while we obtained 6.3, 12.5, and 25 μg/ml for the same. Our results (Table 3) show that the newly designed AMP W3_db5 (KK WKC WAR WRL KK) has an anti-fungal MIC$_{90}$ of 3.75 μg/ml, which is twice better than that of W3_p1 (7.5 μg/ml) and W3_p2 (7.5 μg/ml), while it is a level lower in its bactericidality with a MIC$_{90}$ of 30 μg/ml than that of W3_p1 (15 μg/ml) and W3_p2 (15 μg/ml). The latter result is consistent with the MD-based natural insertion results (Supplementary Fig. 3).

Although our newly designed W3_db5 has similar anti-bacterial and anti-fungal activities as its mother templates', it has surprisingly low cytotoxicity as compared with the template AMPs. As shown in Fig. 4, at the concentration of 240 μg/ml, W3_p2, W3_p1 and W3_db5 lyse 20.3%, 13.0% and 0.6%, respectively, the red blood cells (RBCs). W3_db5 is at least 20-fold less hemolytic than W3_p1 and W3_p2 (at least 12-fold less hemolytic at 120 μg/ml); in other words, we can use 20-fold higher concentration than W3_db5's MIC to still have smaller cytotoxicity than its W3 templates. In Table 3, we also list AMPs' minimal hemolytic concentration (MHC$_5$) to lyse 5% of the RBCs, where W3_db5's MHC$_5$ can be much larger than 240 μg/ml, which implies even a high concentration of W3_db5 can still be potentially used in medication or personal hygiene products.

To further validate the use of TP-DB, as a control study, we borrow the conventional PHI-BLAST to find a sequence homologous to W3_p1 and containing the motif "W2W2W". Here we have to note that PHI-BLAST has to take two inputs to initiate a search—a protein sequence S and a regular expression pattern P occurring in S. Given the two inputs, PHI-BLAST helps finding protein sequences both containing an occurrence of P and are homologous to S in the vicinity of where the pattern P occurs. Here, P as "W-x(2)-W-x(2)-W" and S as the sequence of W3_p1 are used to search the PDB via the NCBI PHI-BLAST website. A single-chain protein (PDB: 4F54) from the DUF4136 family with a relatively high resolution (1.60 Å) contains a stretch of "WGN WGG W", located at the end of a helix connecting toward a beta-sheet through a short loop. We synthesize a peptide "KK WGN WGG WRL KK", coined as "W3_n1", which replaces the "WKC WAR W" part in W3_db5 with the "WGN WGG W". When examining its MIC, we fail to find its MIC$_{90}$ given the concentration range we have tested (the highest at 120 μg/ml) (see Table 3), suggesting not any non-helical "W2W2W" sequence can work as a functional AMP motif. As a second control study, we replace the "WKC WAR W" part in W3_db5 with "WKD WES W", from the 8$^{\text{th}}$ unique sequence in Table 2, to create the peptide "KK WKD WES WRL KK", termed as "W3_n2". The "WKD WES WRL" stretch has one less positively charged residue and two more negatively charged residues (D and E) as compared with the "WKC WAR WRL" motif in W3_db5. Again, we do not find its MIC$_{90}$ within the concentration range we have tested (the highest at 120 μg/ml) (see Table 3), which suggest a helical stretch containing the W2W2W motif does not

necessarily guarantee its bactericidal activity if there are not enough positive charges in the designed AMPs. Note that the TP-DB-found W3_db5 has the same number of positive charges as its W3 templates (i.e., W3_p and W3_p2), while W3_n2 could be 3 positive charges short to become an AMP. In the MIC$_{90}$ experiments, W3_n1 and W3_n2 respectively kill 2.5 and 9.5% of the *E. coli* ATCC 25922 at a concentration of 30 μg/ml, which is the MIC$_{90}$ value of W3_db5 for the same bacteria.

It should be noted that the original protein (mycobacterium tuberculosis Zinc-dependent metalloprotease-1; PDB ID: 3ZUK) that contains this potent stretch "WKC WAR WRL" in W3_db5 is not a transmembrane or membrane-bound protein; the helical stretch is situated close but not fully exposed to the water-protein interface. Therefore, TP-DB brings an interesting opportunity for researchers to put together structurally resolved elements, matching the desired pattern, for the rational design of therapeutics, herein showcased by a new class of AMP that is 6 point mutations away from its original templates—W3_p1 and W3_p2.

From the design point of view, these 6 point mutations do not seem to change much the required insertion thermodynamics[42]. The resulting AMP, W3_db5, can still penetrate the peptidoglycans, reach the lipid-water interface, stay amphiphilic and helical, and eventually insert into the cell membrane. This is difficult to achieve by mutagenesis experiments or MD simulations without the guidance of TP-DB.

**TP-DB-selected residue substitution provides a higher helicity than those derived by other means.** Taking our peptides, we would like to see whether the sequences extracted from TP-DB can be more helical in its isolation than similar ones (sequences with the same pattern) suggested by other methods (e.g., PHI-BLAST). Our circular dichroism (CD) data for W3_db5, W3_n2, and W3_n1, flanked by the same residues in the template (namely, KK … RLKK), showed that the three have different degrees of helicity (Fig. 5), where the "WxxWxxW matches" of the former two are extracted from the TP-DB and that of W3_n1 is taken from the PHI-BLAST search results. We found W3_db5 adopts a higher helicity than W3_n2 that is found to be more helical than W3_n1 (Fig. 5), where W3_n1 has a helicity closest to Tat, a popular cell-penetrating peptide known to take a random coil structure[43]. This low helicity is consistent with the fact that the "WGN WGG W" stretch in W3_n1 takes mostly coil and sheet structures in its original PDB file (PDB ID: 4F54). The online software DichroWeb[44] provides a helicity percentage prediction for Tat, W3_n1, W3_n2, W3_db5, W3_p1, W3_p2, and CM15 based on their spectra data over the wavelength 190 to 260 nm wavelength −6.5, 2.0, 18.3, 60.2, 57.8, 51.2, and 54%, respectively. Interestingly, the helicity percentage has a correlation coefficient of 0.78 with the HP (log(HP$_{NA}$)) of aforementioned peptides −0.406, 0.011, 1.418, 1.675, 2.012, 1.940, and 0.852, respectively.

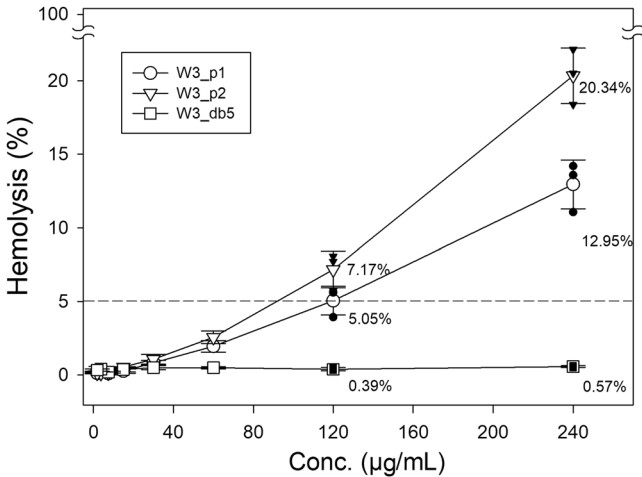

**Fig. 4 The hemolytic percentage as a function of AMP concentrations for W3_p1, W3_p2, and W3_db5.** W3_db5 has much lower hemolysis of red blood cells (RBCs) than that of W3_p1 and W3_p2. The dashed line indicates the 5% hemolytic threshold that defines $MHC_5$. Data are presented by the mean ± S.D. from three repeats. W3_db5 values are statistically smaller than W3_p1 ($p = 0.00021//0.00118//0.0032$ at $240//120//60 \mu g/ml$) and W3_p2 ($p = 0.00005//0.00073//0.00118$ at $240//120//60 \mu g/ml$) according to Student's $t$ test (two-tailed), and the experimental data are provided as a Source data file. Black points at 120 and 240 µg/ml denote the individual data points for the 3 repeats.

### Design peptide inhibitors for tumorigenic PPI.

Our statistics in Table 1 also reveal the interesting fact that alanine is indeed a safe choice for point mutation while still maintaining the integrity of a helix (see $HP_{NA}$). However, with the pattern-based search engine (see "Methods" section "Structural properties for the pattern-matched helical peptides in TP-DB") made available in TP-DB, one can do point mutation(s), by choosing the residues suggested by TP-DB, in a helix to abolish or enhance (see Supplementary Information) its interaction with certain drugs, peptides or proteins while still maintaining the helicity of the helix, as well as to abolish the catalytic function of a residue on an α-helix while maintaining its binding affinity with certain substrate or ligand in the scenario where a structural biologist co-crystalizes an enzyme with its cognate substrate. In the Supplementary Information, we demonstrate an in silico study on how peptide blockers can be designed by TP-DB to block a helix–helix interface between Sgo1 and PP2A.

### Discussion

Zimm–Bragg[45] and Lifson–Roig[46] coil–helix transition models suggested that the propagation of a helix can occur more easily than nucleation of a helix from the coil state. This suggests that, given our helical templates, point mutations (substituted residues) can be more likely in their helical forms than it following a coil structure. In our AMP and anti-tumor peptide design, a pattern with functional importance is first known and then pattern-matched helical stretches are found by TP-DB to suggest structurally "suitable" residue substitutions for the next-round selection to further address the functional need. The anchoring residues (i.e., the three tryptophan in "WxxWxxW") have to be in the found helical stretches and consistently serve as the first neighbors of "x" before and after the stretch replacement (e.g., the "WKC WAR WRL" was taken from "NVED WKC WAR WRL IRARA" in the original protein and substitute the "WRK WLK WLA" part in W3-p1). The H-bond formation between any residue and its third or fourth neighbors throughout the

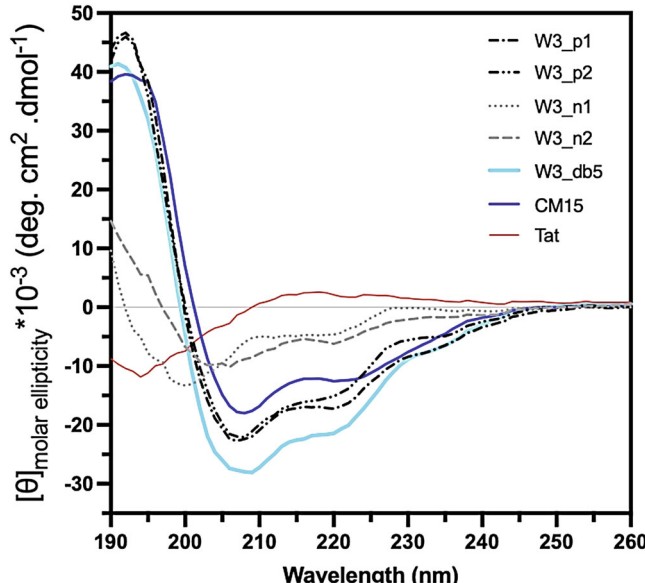

**Fig. 5 The CD spectra from 190 to 260 nm for all the examined peptides in the study where CM15[1,50] is a known helical AMP and Tat, a cell-penetrating peptide (YGRKKRRQRRR) extracted from the transactivating transcriptional activator, is in a random coil structure[43].** All the experimental data are provided as a Source data file.

"WKC WAR WRL" is evidenced in earlier structural determination. On the other hand, whether "WCK WRA WRL", having the same HP as W3_db5, would result in the same helicity is less assured than what was experimentally determined (see discussions on Max_sum4 toward the end of Discussion and Supplementary Table 4). Other choices of "x" according to averaged properties such as HP, without the use of TP-DB, can also possibly result in a helical form. However, such a choice forming the helical structure is still lacking experimental support for their $i - i + 3$ H-bond formation. One example is that poly-tryptophan, even at relatively high concentration (~2 mM), is not necessarily helical in 0.1 M NaCl solution based on previous circular dichroism data[47] despite the tryptophan having the second-highest HP (Table 1). Another advantage of taking helical fragments from living organisms (in TP-DB) instead of artificially synthesized ones (e.g., W3-p1 and W3-p2) can be biocompatibility. Striking hemolysis reduction of human RBCs for W3-db5 by 20–40-folds, as compared to its templates W3-p1 and W3-p2, is inspiring besides being encouraging, although the detailed mechanism cannot be revealed without further studies.

To further quantitatively estimate how helical the TP-DB peptides in their isolation can be, including AMPs, already in isolation, and TP-DB-stored peptides, we conducted the following aiming to provide the helicity percentage estimate as a function of HP, peptide concentration, and average residue contact in peptides' original proteins. Here, helicity percentage is simply defined as the percentage of the residues in a peptide being in the helix form. First, from a PDB search and literature survey, we compiled an "isolated peptide set" comprising 37 NMR-resolved helical peptides with a helicity percentage from 37% to 96% (see Supplementary Table 3 and Supporting Methods for collection details of the set). Here "isolated" is defined as averaged residue contact in a collected PDB structure that should be <7.6 amino acids given a 7.3 Å Cα-Cα cutoff[22], and "helicity percentage" is defined as the percentage of a peptide being helical according to the header information in its PDB file. According to Table 1, one can calculate the $\log(HP_{NA})$ value, HP, for each of these 37 peptides. In Fig. 6a, one can find helicity percentage grows with HP with a

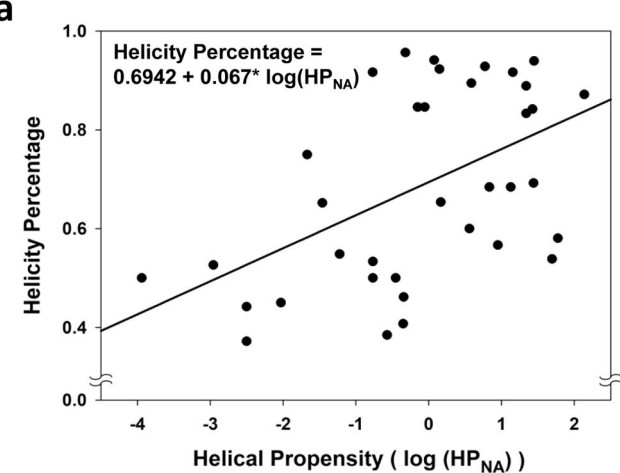

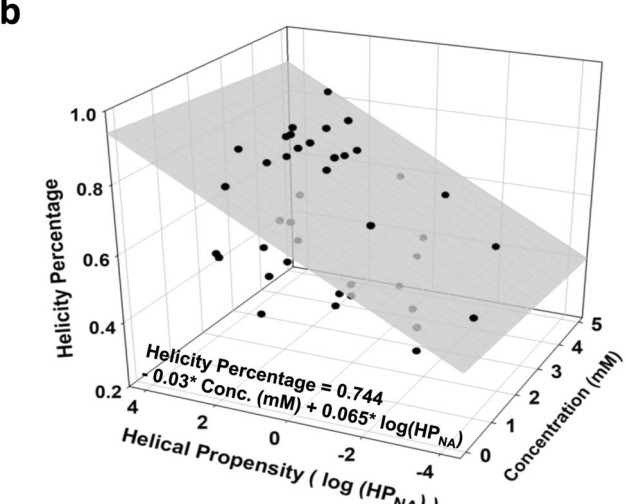

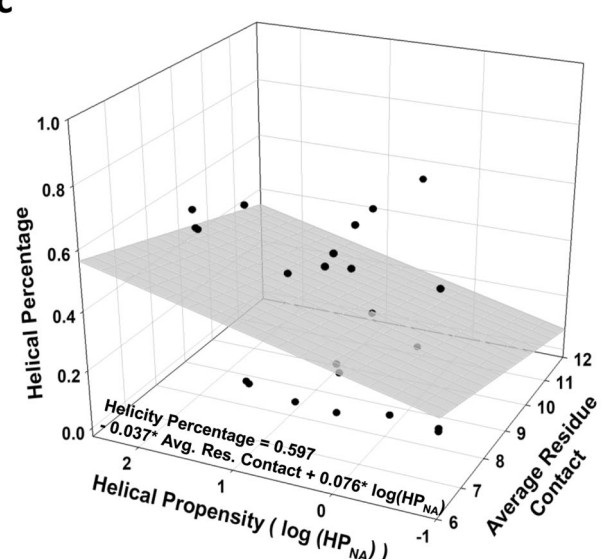

**Fig. 6 Helicity percentage as a function of helical propensity, peptide concentration, and average residue contact for the isolated peptide set (AMPs) and the TP-DB-stored peptides. a** (Up) The helicity percentage (ranging from 0 to 1 in this figure) is plotted against the helical propensity for the 37 AMPs in the isolated peptide set. A relation—helicity percentage = 0.6942 + 0.067 helical_propensity—can be found with a correlation of 0.50. **b** (Middle) Helical propensity is plotted as a function of the peptide concentration and helical propensity for the 37 AMPs in an isolated peptide set. Conducting the linear regression, we can obtain a relation —helicity percentage = 0.744 − 0.03 concentration (in mM) + 0.065 helical_propensity with a Pearson's correlation coefficient of 0.53 and an estimated error of 16.1%. If the 4 AMPs, namely, CM15, W3_p1, W3_p2, and W3_db5, examined in our CD study (Table 3 and Fig. 5) are added to the data, the regression gives a relation such that helicity percentage = 0.693 − 0.009 concentration (in mM) + 0.055 helical_propensity with a declined correlation of 0.45 and a 16.9% estimate error. **c** (Bottom) From the last 50 ns of 100 ns MD trajectories of 23 TP-DB-stored peptides containing no proline, we can derive that Helicity_Percentage = 0.597 − 0.037 × Avg. Res. Contact + 0.076 × Helical_Propensity with a correlation of 0.6, from which the estimated distribution of helicity% for the entire TP-DB can be derived (Supplementary Fig. 2d).

concentration) has a higher helicity percentage than that of the same measured by CD (at a lower concentration). For instance, an intragenic AMP (PDB ID: 6VLA) has a helicity percentage of 94% according to its NMR-resolved structure but 59% by CD[48]. Similarly, an AMP from Whiteleg shrimp (PDB ID: 2N1C) has 65 and 35% helicity percentage when structurally resolved by NMR and CD, respectively[49]. The CM15 we used for control in the CD experiment was previously found to have a 65% helicity percentage (compared to our measurement of 54%) by CD[50] but 80% helicity by NMR[1]. To reflect such a concentration dependency of helicity percentage, we surveyed the concentration of AMPs in their structural determination for our isolated peptide set (Supplementary Table 3). When considering both HP and peptide concentration, we obtained a relation of helicity percentage = 0.744 − 0.03 concentration (in mM) + 0.065 helical_propensity with a Pearson's correlation coefficient of 0.53, slightly higher than the correlation if only $\log(HP_{NA})$ is considered, and an estimated error (r.m.s.d. between experiment and prediction; see details in the footnote of Supplementary Table 5) of 16.1%. If adding another 4 AMPs, namely CM15, W3_p1, W3_p2, and W3_db5, tested in our CD study (Fig. 5), we could obtain a linear regression relationship—helicity percentage = 0.693 − 0.009 concentration (in mM) + 0.055 helical_propensity with a fitting correlation of 0.45 and an error estimate of 17.4%. A month after we constructed this two-variable model, we found that two extra NMR-determined isolated peptides also belonged to the "isolated peptide set" and had an averaged prediction error of 13.4%, which provides a further reference for our model accuracy.

For the above analyses, we did not take the average residue contact into account because all these AMPs were already in isolation when they were structurally solved and had an average contact ranging from 6.0 to 7.2 (95% of the data). We also noted that the isolated peptide set did not exhibit expected dependency between helicity % and concentration, evidenced by the −0.03 coefficient for concentration. However, as only 4 CD-measured AMP data were included in the correlation, the coefficient started to grow toward zero (−0.009). In fact, if another three non-AMP peptides (tat, W3_n1, and W3_n2) were included, the pre-concentration coefficient would become +0.043. Given the unbalanced data where our equations resulted from more NMR-than CD-solved structures, these equations could be more suitable to predict helicity% observed by NMR experiments.

On the other hand, we are also interested in the helicity percentage for isolated TP-DB peptides. After deriving the general

correlation of 0.50 and the linear regression provides a relation helicity percentage = 0.067 helical_propensity + 0.6942.

It should be carefully noted that helicity percentage is also concentration-dependent. NMR-resolved structures are usually determined at a concentration of several mM, while CD spectra measure at a few tens of μM. It has been reported by several studies that the same helical peptide resolved by NMR (at high

statistics for parameters including HP and residue contact of peptides in TP-DB (Supplementary Fig. 2a–c), we realized that the average contact is mostly populated in the range of 8.0 to 8.25 (Supplementary Fig. 2b) and HP (Supplementary Fig. 2a) in the range of 0.2 to 0.4. At the most populated $\log(HP_{NA})$, we chose 8 TP-DB peptides having an average residue contact that is within −2 S.D. lower and +4 S.D. higher than the average value (0.89). Together with this, another 15 peptides were chosen to have a contact fixed at 8.0 to 8.25 and $\log(HP_{NA})$ values ranging from −2 S.D. lower and +2 S.D. higher than the average value (0.38). The 23 TP-DB peptides, each of which has a length of 13 or 14 residues, are detailed in Supplementary Table 4 and the Supporting Methods. We purposely did not choose any peptide containing proline that is known to be an effective alpha-helix breaker in non-membrane environment[51] and have the lowest $\log(HP_{NA})$ in Table 1. We then conducted a 100 ns MD simulation for each of the 23 peptides at 100 mM neutralizing ions, body temperature, and 1 atm. The secondary structure assignment for the equilibrated 50–100 ns trajectories was analyzed by VMD STRIDE plug-in[52] to report the averaged helicity percentage (over the snapshots in the latter half of the simulation) for each peptide. Our linear regression showed that Helicity Percentage = 0.597 − 0.037 × Avg. Res. Contact + 0.076 × Helical Propensity (Fig. 6c) with a Pearson correlation coefficient of 0.6. When separately examining the 8 peptides that share similar HP and the 15 peptides sharing a nearly identical average residue contact, we revealed a negative correlation of −0.57 and a positive correlation of +0.51 for helicity% vs contact as well as helicity% vs HP, respectively. However, we should still note that the small sample size presented herein could limit the model generalizability.

In the above regression models, the coefficients before HP for the isolated peptide set and TP-DB peptide set are +0.067 and +0.076, respectively, agreeing with our expectation that observed helicity% grows with $\log(HP_{NA})$. In a similar vein, the coefficient before average residue contact is −0.037, which also agrees with our earlier surmise that higher tertiary contact in peptides' original protein environment may lead to a slightly lower helicity% in their isolation.

Also, the aforementioned HP score sums the log-odd probability of each residue in the peptide (Table 1) regardless of their order in the sequence or the likelihood of forming hydrogen bonds. As a result, one may question how different helicity% can be for peptides with the same HP. We designed such a study and found that peptides with the same amino acid composition in randomly shuffled order indeed can have quite distinct helicity% (bottom of Supplementary Table 4). Considering the coil-helix transition models[45,46], we anticipate that helicity is easier to maintain if a structured nucleus comprising 3 to 4 consecutive residues in the sequence can have a higher sum of the HP, which may help reserve the backbone hydrogen bonds for TP-DB peptides in isolation. We thus defined a term, Max_sum4, that is the highest sum of $\log(HP_{NA})$ values of all the 4 residues among all the 4-residue windows in a peptide, representing the strongest folding nucleus in that sequence, and examined the correlation between Max_sum4 and MD-revealed helicity percentage. We indeed found a correlation of 0.63 when examining the "wide type" + 5 mutants and 0.73 for the 5 mutants (Supplementary Table 4), which could inspire a further examination of the usefulness of such a property for a larger set of isolated peptides in the future.

Although we demonstrated that W2W2W motif can be presented as part of a good AMP design template, there are factors AMP designers should pay attention to, including but not limited to sufficient positive charges and HP (e.g., W3_db5 are higher than W3_n2 in both of these factors), in addition to the fact that the W2W2W motif can already be flanked by residues in their helical form and/or other positively charged residues in the N/C-termini.

To conclude, the pattern-based (without relying on sequence homology) and BLAST-based (relying only on sequence homology) search engines for the protein sequences that have been observed to have a specific type of secondary structures, herein α-helices, were established and implemented online with efficiency to find pattern-matched sequences from 1.67 million peptides within usually half a minute. With these search engines, to introduce mutations for gain of function while securing a desired secondary structure was made possible for the first time. With such a facility, we introduced how one can utilize our database to design a new helical AMP that was later verified to be anti-fungal and anti-bacterial, while its high helicity% was verified by circular dichroism spectra. We also demonstrated that one can use this database to design helical-peptide blockers to prevent a tumorigenic protein–protein interaction. Improved therapeutics can therefore be designed not only with known functional sequence motifs but also with desired secondary structures, while the helicity% for our TP-DB in isolation, if needed, can be estimated from our parameters including HP and average residue contact. We also foresee its future extension to incorporate search engines for beta-sheet elements containing specific sequences and/or patterns to meet desired physico-chemical traits with a design purpose.

## Methods

**Extraction of helical peptide sequences from the PDB.** To obtain the amino acid sequences that fold into helices in nature, we processed the PDB[23] files of 130,000+ experimentally determined protein structures as of June 2017 and extracted the locations of secondary structures from the header part of the PDB files. There are 96,711 unique PDB IDs/files containing at least one alpha-helix documented at the header area to result in 1,676,087 structurally resolved helices having a length longer than 5 residues (one residue more than the number required to form an α-helical pitch). In addition, for a given helix, its contacting neighbors in 3D space and the adjacent helical peptides that are within 4 Å (per heavy atoms) were also collected and analyzed.

**Structural properties for the pattern-matched helical peptides in TP-DB.** The helical peptides returned by a query can be ranked by (the logarithm of) their normalized HP (default) or by their contact scores. Using the data collected in the TP-DB, each of the 20 amino acids has a normalized HP, HPNA that is $HP_{a.a.}/NA_{a.a.}$, where $HP_{a.a.}$ is the HP of a given amino acid type and $NA_{a.a.}$ is natural abundance of that amino acid (Table 1). Both $HP_{a.a.}$ and $NA_{a.a.}$ are probabilities, where $HP_{a.a.}$ is defined as the count of a certain amino acid type in all the extracted helices divided by the total number of amino acids in these helices (24,301,682 amino acids in 1,676,117 helical peptides), and $NA_{a.a.}$ is the count of that amino acid in the 130,000+ proteins in PDB, divided by the total number of residues in these proteins. A log-odds ratio, $\log(HP_{NA})$ or $\log(HP_{a.a.}/NA_{a.a.})$ can be found for each amino acid type, and the reported HP for each peptide in TP-DB is the sum of $\log(HP_{NA})$ for every constituent residue in this peptide (Table 2). The contact number is the amino acid contact within a 7.3 Å range of a residue of interest, in the residue's original protein environment, where only the Cα atom of an amino acid is considered. The reported contact for a peptide in TP-DB is the per-residue average of its constituent residues. An exemplified output table from the webserver is shown in Table 2. We would like to note that the HP and contact are calculated only for the "matched part" (e.g., "WKCWARW" in Table 2), not the corresponding full helix (e.g., "NVEDWKCWARWRLIRARA" in Table 2) stored. Because it is impossible to know beforehand what users will search, the two quantities are calculated on the fly, not originally stored in the database.

**Lipid preparation and equilibrium in silico.** Zwitterionic membranes comprising 30 1-palmitoyl-2-oleoyl-sn-glycero-3-phosphocholine (POPC; also termed as "PC" hereafter) lipid and 10 2-oleoyl-1-pamlitoyl-sn-glyecro-3-glycerol (POPG; also termed as "PG") lipid molecules (see Supplementary Fig. 1) in each of the upper and lower lipid bilayers to mimic pathogenic prokaryotic membrane[28,29] were prepared by CHARMM-GUI server[53]. The AMPs and lipid bilayers were solvated in explicit TIP3 water molecules[54]. Neutralizing ions were added to maintain a concentration of 100 mM. The initial area per lipid of PC was equilibrated to stay close to its experimental values measured at 303 K, which is 68.3 ± 1.5 Å[2 55].

**MD simulations of TP-DB peptides and AMPs**. Unless otherwise stated, all MD simulations were performed using either NAMD package 2.9[56] or OpenMM[57] with CHARMM36 forcefields[58,59]. Simulations were performed for 100 ns and 1 μs in the case of TP-DB peptides and AMPs, respectively, at a time step of 2 fs. RATTLE and SETTLE algorithms are applied to constrain hydrogen atoms in peptides and waters. Cutoff of 12 Å with switch distance 10 Å and pair list distance 14 Å are applied when calculating non-bonded interactions. With periodic boundary conditions, the Particle Mesh Ewald method[60] was employed for calculations of long-range electrostatic interactions. The temperature was maintained at 310 K using Langevin dynamics[61] and pressure was controlled at 1 atm using Nosé-Hoover Langevin piston[62] or Monte Carlo Barostat for the peptides in the water box with 100 mM neutralizing ions. Energy minimization and NVT heating were carried out with a weak restraint on Cα atoms before the production runs in NPT without any constraint. For AMPs in the presence of aforementioned PC/PG (3:1) membranes, A soft-boundary condition is applied to prevent peptides from leaving the boundary that is 35 Å above the center of the membrane by exerting a force that is a small force constant 3 kcal/mol/Å$^2$ multiplied by the distance exceeding that boundary[2].

**Determination of the MIC**. To measure the antimicrobial activity of peptides, MICs were determined by the broth-dilution assay as described in Clinical and Laboratory Standards Institute (CLSI) documents M27-A3 for yeasts[63] and M07-A9 for bacteria[64] with minor modifications. Briefly, E. coli ATCC 25922 and C. albicans SC5314 were recovered from frozen stock by growing overnight at 37 °C on LB (Luria-Bertani) and YPD (Yeast Extract-Peptone-Dextrose) agar plates, separately. A single colony of E. coli was inoculated into 5 ml LB medium and grown overnight at 37 °C with shaking at 220 rpm. Cells were then diluted into MHB (Mueller–Hinton broth) at 1:100, and grew until the optical density at 600 nm ($OD_{600}$) reached 0.4 to 0.8. For C. albicans SC5314, a single colony was inoculated in YPD broth and grown overnight at 30 °C with shaking at 180 rpm. Cells from the overnight culture were subcultured into fresh YPD and grew to mid-log phase. Finally, E. coli and C. albicans cells obtained from the above-described were diluted into MHB and LYM, a modified RPMI 1640 medium[65] to a concentration of ~5 × 10^5 cells/ml and ~4 × 10^4 cells/ml, respectively. To determine the MIC values, one hundred microliters of cells was placed in each well of a clear bottom 96-well microplate, followed by independently adding 100 μl peptides with different concentrations (in MHB or LYM). The final concentrations in the wells were 120, 60, 30, 15, 7.5, 3.75, and 1.88 μg/ml for peptides, ~2.5 × 10^5 cells/ml for E. coli and ~2 × 10^4 cells/ml for C. albicans. The plate was incubated at 37 °C with shaking at 220 rpm for 18 h (E. coli) and at 180 rpm for 24 h (C. albicans). The $OD_{600}$ values were measured using a iMark$^{TM}$ Microplate Absorbance Reader (Bio-Rad). The $MIC_{90}$ value was determined by 90% reduction in growth compared with that of the peptide-free growth control. Experiments were performed in pentaplicate, and $MIC_{90}$ was determined as the majority value out of 5 repeats.

**Assays for minimum hemolytic concentration (MHC)**. The hemolytic toxicity was determined by the hemolysis against human RBCs as in previous publication[66]. First, RBCs from healthy donors were mixed with EDTA in the collecting tubes to inhibit the coagulation. Then the RBCs were washed for three repeats as follows—centrifugation at RCF 800 × g for 10 min, removal of the supernatant, and resuspension with PBS buffer. Such washed RBCs stock was finally diluted with a PBS buffer to be the 10% (v/v) RBCs solution for assays. Then total eight samples of various final concentrations (240, 120, 60, 30, 15, 7.5, 3.75, and 1.88 μg/ml) for each AMP in 96-well microplate were prepared by mixing 10% RBCs and diluted AMP solutions, respectively, at ratio 1:1 (v/v) into final volume 200 μl. All samples were placed under the condition of 37 °C for 1 h and then centrifuged at RCF 800 g for 10 min. Finally, all these supernatants from samples and controls were picked out for the measurement of optical density at the 450 nm wavelength. The data processing was performed to calculate the hemolytic percentage defined as $[A_{sample} − A_{control−}]/[A_{control+} − A_{control−}]$, where $A_{control+}$ and $A_{control−}$ are the measurement at wavelength 450 nm for the positive and negative controls, that are 100 μl 10% RBCs suspended in 100 μl PBS buffer with and without 2% (v/v) Triton X-100 (a known detergent to lyse the RBCs), respectively. The final hemolytic percentage for AMPs at various concentrations was determined by the mean values of three repeated experiments, and $MHC_5$ value, defined as the minimal concentration of AMPs leading to at least 5% hemolysis of RBCs, can be determined.

**Ethical statement**. The hemolysis protocol was pre-approved by the Research Ethics Committee of the National Taiwan University Hospital (approval number: 201810004RINA). The blood collections were performed by trained phlebotomists to confirm the safety of the donors. The relevant informed consents were provided to the donors to obtain their agreement on this experiment prior to the process, and the nutritional supplements as compensation for research participants were also provided after the operation.

**Reporting summary**. Further information on research design is available in the Nature Research Reporting Summary linked to this article.

## Data availability

The authors declare that the data of TP-DB are available at GitHub (https://doi.org/10.5281/zenodo.5675428)[67] and Zenodo (https://doi.org/10.5281/zenodo.5653287)[68]. Main data supporting the findings of this study are available in the Supplementary Information, Source Data file, and other Supplementary Data, including Supplementary Movie 1 and Supplementary Data 1. Supplementary Movie 1 describes the insertion process of AMPs into the POPC/POPG (3:1) membrane observed in MD simulations and is available in both Nat. Commun. website (Supplementary_Movie_1.mov of 18.9 MB) and Zenodo repository (Supplementary_Movie_1.mp4 of 76.3 MB with higher resolution). Supplementary Data 1 containing all the helices used in this study is made available in the Zenodo repository (file name "Supplementary_Data_1.zip"). The corresponding descriptions of each helix, including the sequences, their sources in PDB, and hyperlinks to their structure files, are also available in the Zenodo repository (see the README file on the Zenodo repository). The third-party image shown in Supplementary Fig. 4 is reproduced from the review article (Marston, A. L. "Shugoshins: tension-sensitive pericentromeric adapters safeguarding chromosome segregation." Mol. Cell. Biol. 35, 634–648 (2015). Its reuse permission (permission for the reuse of Supplementary Fig. 4.pdf) is obtained through the Copyright Clearance Center of the publisher (https://marketplace.copyright.com/rs-ui-web/mp). Extra data are available from the corresponding author upon reasonable request. Source data are provided with this paper.

## Code availability

The executable source codes of the search engine for TP-DB are available at GitHub (https://doi.org/10.5281/zenodo.5675428)[67] and Zenodo (https://doi.org/10.5281/zenodo.5653287)[68] repositories.

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

## Acknowledgements

We acknowledge Dr. Danny Sheng-Te Hsu for helpful discussions. E.O.S. acknowledges financial support from MOST and Taiwan International Graduate Program, Academia Sinica, Taipei, Taiwan. We thank vast computational resources for this project provided by High Performance Computing Infrastructure (HPCI), Japan and the National Center for High-performance Computing (NCHC) of National Applied Research Laboratories (NARLabs) of Taiwan. This work was supported by the Ministry of Science and Technology, Taiwan (103-2627-M-007-001 and 104-2113-M-007-019 to L.-W.Y.; 104-2311-B-007-003 and 108-2311-B-007-004 to H.-W.F.; 105-2311-B-007-007-MY3 and 109-2311-B-007-001-MY3 to C.-Y.L.), the research program of National Tsing Hua

University (to H.-W.F.), and the National Natural Science Foundation of China (22007097 to H.L.).

## Author contributions

L.-W.Y. conceived and designed this study; C.-Y.T., G.-Y.L. B.-S.F., and C.-Y.L. jointly designed the protocols of MIC tests for AMPs/AFPs, where the experiments were conducted by C.-Y.T., L.V., K.-D.H., S.K., Y.Y. Cheng, and S.S. C.Y.T. and C.W.C. performed the hemolysis assays and analyzed the data with the help of C.-C.W. C.-Y.T. and E.O.S. designed the anticancer peptides. Y.Y. Cheng and C.-Y.T. designed and conducted CD experiments. Y.Y. Chang collected 1.67 million helices with the help of A.S. E.O.S. gathered the helix data and built the TP-DB database, designed the algorithm of search engines, and analyzed the data; E.O.S., H.L., Y.Y. Chang, and Y.-L.L. jointly constructed the website interface of TP-DB. H.L. performed the MD simulations for isolated TP-DB peptides and analyzed the data with the help of L.S. and C.L.D.O.; G.-Y.L. designed and performed the MIC assays with L.V. and K.-D.H.; T.-Y.K. performed functional assays of anti-FLAG M2 antibody against the HPNAP supervised by H.-W.F.; L.V. performed the MD simulations of AMPs/AFPs in PC/PG lipid membrane and analyzed the data with the help of L.S. L.-W.Y., E.O.S., and C.-Y.T. drafted the manuscript that is co-edited by H.L. and H.-W.F.

## Competing interests

The authors declare no competing interests.
