## [Peer Review File · Nature Communications]

Helical Structure Motifs Made Searchable for Functional Peptide DesignReviewers' Comments:

Reviewer #1:

Remarks to the Author:

In this paper, Tsai et al. presented a novel 3D secondary structure motif database called Therapeutic Peptide Design database (TP-DB) for functional peptide design. The database contains curated alpha helical structures retrieved from PDB made searchable using compounded patterns. Specifically, they developed two parameters helical propensity and contact scores and showed that these two parameters could better prioritize helical structures than common homologous search. They demonstrated the utility of TP-DB by designing a new antimicrobial peptide with improved potency and lower toxicity. In another application, they showed in silico how TP-DB can suggest point mutations on helical structures to improve peptide-peptide interaction in the development of tumor suppressor. Overall, I feel quite positive about the paper based on the extensive in silico and experimental efforts in support of their finding and the uniqueness of their 3D secondary database, which fill an important gap in the field. However, I think several points should be addressed to make their work of broader interest to the audience.

1. I would suggest making the extracted helical 3D structures (PDB coordinates) available for download by clicking on the sequence as this will help users evaluate the retrieved helical conformations as well as for subsequent peptide design.
2. The authors developed two parameters: helical propensity score and contact number score to predict if retrieved sequences from PDB could maintain helical structures in isolation. Is there an optimal cutoff for the scores and what is the error rate like? These parameters should be more rigorously evaluated to help users better select peptides from the search.
3. Is there any correlation between helical propensity score and contact score? If I were to use these two parameters to prioritize the peptides, which should I look at first? If they are independent, I think the use case of each should be more clearly defined. Another alternative is to develop a combined score based on these parameters which would be easier for the user.
4. Line 240, "TP currently comprises 1,676,117 helical peptides.." How many peptides in the database were (expected to be) helical and stable in isolation? I would also recommend providing additional statistical profiling and visualization of the database.
5. More validation is needed to demonstrate the utility of helical propensity scores and contact number scores for inferring helical conformation and stability accurately. My concern is that the helical propensity score simply sums the log-odd probability of each residue in the peptide regardless of their order in the sequence or the likelihood of forming hydrogen bonding. Similarly, a high contact score does not necessarily indicate that the helical structure will not be stable. I would suggest performing MD simulation on selected helical sequences with high vs low helical propensity score/contact numbers and then compare their conformation/stability over time.
6. Similarly, for section 3.4, perform MD simulation comparing peptides with point mutations using residues with better/worse helical propensity scores (Table 1) and show that mutant helical conformation could become more stable or unstable than the wild type.
7. Line 137, I don't think helical propensity or contact scores could predict physiochemical properties without utilizing additional chemical descriptors. If the author means otherwise, the term should be more clearly defined.
8. In Section 3.3, the authors identified antimicrobial peptide (AMP) search pattern "W**W**W" by visually inspecting the MD trajectory (line 339-392) which is quite impressive. To complement this, I am wondering if similar search patterns could be more quantitatively determined, for example, by profiling the non-bonded energy of peptide residues based on the MD trajectory, as performed in figure S4 and S5?
9. Line 332-333, is there any difference in the contact mode between W3_p1 and W3_p2?
10. The authors described a newly designed AMP W3_db5 (KK WKC WAR WRL KK) and several others that had better antimicrobial activity and lower toxicity (Figure 4 and Table 3). However, it is unclear how these improved peptides could be prioritized or designed based on the established parameters e.g. helical propensity or contact scores. Likewise, in Fig. S5, they showed several mutants with

improved binding to Sgo1 but did not explain if their binding activity can be correlated to these two parameters (better helical conformation for binding due to higher helical propensity mutation, for example). To demonstrate a wider applicability of their database, I think it is critical to establish some correlations/connections between the observed experimental finding and the proposed parameters in TP-DB.

11. Along with point 9, I found applications of TP-DB such as those described in section 3.4 for Sgo1 and PP2A binding and elsewhere in the paper rely heavily on MD simulation to assist in peptide prioritization. From the user's standpoint, using the database could be challenging if MD simulation is needed to reduce the candidate pool. Ideally, the authors should demonstrate that TP-DB could perform peptide prioritization independently or implement wider ranges of descriptors in the database to help users characterize the peptides from the database directly.

12. The conclusion is quite brief and should be improved. The authors should summarize what has been done and their unique contribution to the field and how their work could be extended and applied for further peptide design.

Minor comments:

1. Figure S2, missing y-label (relative position).

2. The definition should be given for amino acid abbreviation W (Tryptophan) and Y (Tyrosine).

3. Line 64, define DSSP.

Reviewer #2:

Remarks to the Author:

Overall, the goal of the work is interesting, however as delineated in detail below it does not represent the kind of contribution expected for Nature Communications.

Specific comments:

- Most AFPs and AMPs target the membrane, and the tool presented does not cover these cases. Therefore, the title of the paper must be changed to something more specific.
- Unclear how this database can be used for the design of helices that do not target specific proteins.

Abstract:

- Wording is confusing. The authors need to reword the following: "By our pattern-based search engine but not PHI-BLAST, we identify a motif DYKYLE recognized by the anti-FLAG M2 antibody and repurpose a known purification-tag-specific antibody into a diagnostic kit for *H. pylori*. Also with TP-DB, we can design a new antimicrobial peptide (AMP) that contain the MD-elucidated membrane-insertion pattern WXXWXXW. The AMP has a better minimal inhibitory concentration and a much lower cytotoxicity against *Candida albicans* (fungus) than its template homologs."

Introduction:

- The secondary structures were oversimplified, with an exclusive focus on alpha-helical structures. Thus, it would be more accurate to change the title accordingly (perhaps to Helical motifs database for functional peptide design).
- The authors should reword the following for clarity: "Take amphiphilic antimicrobial peptides (AMPs) for example, if the i -th position is a membrane-insertion-promoting amino acid (say, W or Y), the $i+3$ or $i+4$ -th position needs to be an amino acid also prompt for membrane insertion (say, another hydrophobic residues)." Specific examples should be selected, as opposed to generalization beyond what the data is showing. There are several families of twisted helices (polar angle varies from helical step to the next helical step), which are active. Those peptides, for instance, present scrambled hydrophobic/hydrophilic faces' interface if projected in a helical wheel.

Methods:

- What was the rationale behind selecting 5-residue long sequences? At a minimum, 7-8 amino acids

are needed to adopt helices due to hydrogen bond formation.

- Unclear how the authors account for neighbor residues effects, which are extremely important for the design of helices.
- The authors should not use helical propensity (i.e., helical penalty) to indicate frequency of amino acid residues in naturally occurring helices. Helical propensity is a thermodynamical parameter extracted through elaborate calorimetric measurements in different solvents/mixture of solvents showing the propensity in kcal/mol of a residue to start a helical step given both the most proximate amino acid residue neighbors.
- Are 2 fs and 5,000 steps enough for the MD simulations of secondary structures? Provide references to justify this selection.
- Authors should justify why they used POPG to mimic bacterial membranes if Gram-negatives are mostly composed of PE (~67%).
- The authors should specify the blood type used for hemolysis assays. If different peptides were tested against different blood types, experiments should be standardized.

Discussion:

- Unclear if the features defined by the authors describe segments of peptides from natural proteins or for any given amphipathic sequence. As most amphipathic (cationic or not) peptides follow the helix-coil transition theory (Lifson-Roig's theory), it is important to clarify this point.
- The authors should elaborate on the contact number as a feature.
- Topics 3.2 and 3.2.1 are disconnected from the rest of the text.
- Why were W3_p1 and W3_p2 chosen among all Trp-containing peptides? Please add details on why it is important to have the Trp residues equally spaced.
- The authors cannot generalize the discussion for the W2W2W motif since the number of peptides analyzed is too low to draw this kind of conclusions. Authors should tone down their statements.

Reviewer #3:

Remarks to the Author:

Dear colleagues,

This is a well-written manuscript focusing on the discovery of peptide functional motifs with tendency to fold into a α -helix, using the Protein Data Bank (PDB) to create a therapeutic peptide design data base (TP-DB) that can be searched for potential helical therapeutics.

The function of the database query that results from this work will be very useful to other researchers in accelerating their therapeutic research.

While there are no major concerns, it is worth noting that there is no in vivo experiments to validate the claim that the test peptides resulting from the query reported in the manuscript are of therapeutic value. As shown by thousands of AMPs in the literature, in vitro activity usually does not correlate with in vivo efficacy. Therefore, it is difficult (at best) to conclude that the peptide is an effective antibacterial or antifungal solely due to in vitro activity.

Another concern is that the authors present no biophysical evidence that the test peptides identified as helical from the therapeutic search engine are truly helical. Everything is done in silico except for the in vitro functions.

Lines 49-52 and 53-56 require references

Line 64: DSSP needs to be defined.

Otherwise, this has the potential to be very impactful not just for the AMP field but for noninfectious therapeutic applications as well.

Reviewer #1 (Remarks to the Author):

The authors have adequately addressed most of my comments and the manuscript has been much improved. Helical 3D structures have also been added to the webserver with improved functionality and visualization. Below are a few additional comments regarding the new multiple regression model in the revision:

We sincerely appreciate Reviewer 1's further suggestions that have help corroborate our arguments and refine the presentation.

1. More rigorous mathematical description as well as implementation details of the multiple linear regression model, including formal statistical analysis/error estimation should be provided in the method.

Thanks for pointing this out. We add the method associated with this in the SI.

Least-squares fitting to find helicity% as functions of helical propensity, concentration and tertiary contact

Let $\vec{x}_i = [1, x_1^i, x_2^i]$ where x_1 is the helical propensity and x_2 is either concentration or tertiary contact, where i is the index for individual peptides. The helicity percentage y_i can be expressed as a dot product of the independent variable \vec{x}_i and their parameters $\vec{\beta} = [\beta_0, \beta_1, \beta_2]$, where β_1 is the coefficient of x_1 and β_2 is the coefficient of x_2 . We can write

$$y_i \approx \sum_{j=0}^2 \beta_j \times x_j^i = \vec{\beta} \cdot \vec{x}_i \quad (\text{S1})$$

such that the experimentally observed or MD-determined helicity% on the left of the equal sign can be approximated by the dot product on the right. Here, $i=1$ to 37 or 41 for the AMP case and $i=1$ to 23 for isolated TP-DB peptides and therefore the n in the **Eq. S2** below is 37, 41 or 23.

In the least-squares fitting (Rencher et al., 2012), the optimum parameters $\vec{\beta}$ can be obtained from the minimized sum of mean squared loss $\sum_{i=1}^n (\vec{\beta} \cdot \vec{x}_i - y_i)^2$ such that

$$\vec{\beta} = \text{arg}_{\vec{\beta}} \min \sum_{i=1}^n (\vec{\beta} \cdot \vec{x}_i - y_i)^2 = (\mathbf{X}^T \mathbf{X})^{-1} \mathbf{X}^T \mathbf{Y} \quad (\text{S2})$$

where \mathbf{X} is a $n \times 3$ matrix constituted with the helical propensity and concentration (or tertiary contact) of n peptides and \mathbf{Y} is a $n \times 1$ matrix comprising n data points of experimentally observed or MD-determined helicity%. The matrix calculations were carried out using the programming language MATLAB.

The error estimate is further explained in Q4.

2. line 626, the authors noted that the peptide resolved at higher concentration (by NMR) has a higher % helical percentage but on line 635 and 640 the coefficient for the concentration is negative (-0.03).

We appreciate the careful reading. In our data set the average helicity% for the 37 NMR-determined short peptide structures is 0.687 and that for the 4 AMPs (namely CM15, W3_db5, W3_p1 and W3_p2) measured by CD in our study is 0.558, while the average concentration of the 37 peptides is 1.584 mM and that for our 4 AMPs is 0.06 mM. Collectively, we indeed see in average the NMR-solved structures of isolated peptides have a higher helicity% at a higher concentration (25 fold higher than that used in our CD experiments). However, as the Reviewer kindly noted that we do not see the concentration in mM range positively correlated with the helicity% (the coefficient for concentration is -0.03) when all the peptides used are structurally solved by NMR. We start to see that coefficient start to move toward the positive side (-0.009) while still being smaller than zero. Actually the coefficient becomes +0.043 if another three non-AMP peptides (namely, Tat, W3_n1 and W3_n2) we used in the CD experiment are included. Considering the unbalanced quantity between NMR-solved and CD-solved structures of the isolated peptides, we would like to keep the

current regression results while noting that the equations are more suitable for isolated peptides of which the helicity% is examined by NMR. We added the following text with appreciating this limitation (before Figure 6).

“We also noted that the isolated peptide set does not exhibit expected dependency between helicity % and concentration, evidenced by the -0.03 coefficient for concentration. However, as only 4 CD-measured AMP data are included the correlation starts to grow toward the positive end (-0.009). In fact, if another three non-AMP peptides (tat, W3_n1 and W3_n2) are included, the pre-concentration coefficient becomes +0.043. Given the unbalanced data where our equations resulted from NMR-solved structures, these equations are more suitable to predict helicity% observed by NMR experiments.”

3. Could the authors comment on the relative importance of helical propensity v. contact score v. concentration for determining helical percentage in the proposed models? Are the coefficients significant and interpretable?

Thank the Reviewer's comment. We have addressed the concentration issue in the Q2 above. We see the helicity% for isolated helices derived from TP-DB could have a general trend. Along with that general trend, the sign and magnitude of coefficients for helical propensity and contact score are interpretable and agree with our hypothesis. We added a sentence in the Discussion (before Caveat).

“In the above regression models, the coefficients before helical propensity for the isolated peptide set and TP-DB peptide set are +0.067 and +0.076, respectively, agreeing with our expectation that observed helicity% grows with $\log(HP_{NA})$. In similar vein, the coefficients before average residue contact is -0.037, which also agrees with our earlier surmise that higher helical contact in peptides' original protein environment may lead to a slightly lower helicity% in their isolation.”

4. Line 638-641, it is not clear to me how the error rate was determined (%). Was the error assessed by how well the model fitted the existing data or how well the model predicted new peptides not used for fitting?

By "how well the model fitted the existing data". We now add the definition of r.m.s.d in the footnote of the new Table S12 and refer readers to the **details in the footnote of Table S12**, when we mentioned the 16.1% estimated error.

5. The final model for the TB-DP database should be validated. If tested on the new data, the authors could show a scatter plot of predicted v. experimental values for helical percentage.

We thank the good question. To address this, we did the same search as how we have collected the “isolated peptide set” except we required the date to be after Sept 1st, 2021 (before which we collected similar data to construct our regression model). We found only two peptides that are shorter than 30 amino acids containing helices (but not 100% helical) as well as containing neither disulfide bonds nor other materials (including nanoparticles) to stabilize their helices. The two isolated peptides are listed in Table S12 and they have an estimated error of 13.4%, which is slightly lower than the estimated error shown in Figure 6. We added the following sentences before Figure 6 –

“A month after we constructed this two-variable model, we found two extra NMR-determined isolated peptides that could also belong to the “isolated peptide set”, through which we could validate the model with an estimated error of 13.4%.”

6. The linear model for the TP-DB was fitted using only 23 peptides but then used to estimate the helical percentages for the 1.7 million peptides Fig. S2(d). Although the training data seems well-sample, the author should comment if small sample size and sequence variability could limit model generalizability.

Yes, indeed. We have acknowledged the limitation as the Reviewer suggested in the section on isolated TP-DB peptides in Discussion by adding a sentence “**However, we should still note that the small sample size presented herein could limit the model generalizability.**”.

7. Line 700, was the new descriptors Max_sum4 fitted along with other parameters (e.g. helical propensity, contact number, concentration) in the linear model? The equation and coefficients should be provided.

We fitted it alone not with other parameters. The new equation has been added to the footnote of Table S11.
(Helicity% = 3.70 Max_sum4 - 2.82)

8. The Max_sum4 descriptor with sliding windows appears to be promising. To tackle sequential data, have the authors also look into the applicability of deep learning frameworks such as CNN or RNN?

We well appreciate the suggestion. The 4-gram feature can be of use to the suggested AI tests in similar studies. However, in our case, it can be somewhat challenging simply because the amount of labeled data (namely, helicity% for solvated short helical peptides in isolation) are too few (we had only 37 peptides in Table S10) for us to perform a good NN model.

Minor comments -

1. Fig. S2D, missing x label, helical percentage (%), y label counts(N)

Now both x and y labels are added in the Fig S2D as follows:

2. Fig. S2 line 52, NA (also elsewhere) should be subscripted.

Much thanks for the correction. After inspecting the whole content of main-text and supplementary material, two “NA” are revised to be subscripted in the caption of Fig. S2 and Table S11 as follows:

“(d) The estimated helicity % for TP-DB, excluding proline-contained peptides. The estimation is based on the relation Helicity Percentage = 0.597 - 0.037 * Avg. Res. Contact + 0.076* $\log(\text{HP}_{\text{NA}})$, established by MD simulation results of 23 TP-DB peptides (see Fig 6).”

“Residues underlined are the location of folding cores, having the highest sum of $\log(\text{HP}_{\text{NA}})$ values (Max_sum4) among all the 4-residue windows in that peptide.”

3. line 306-307, "to our heartfelt delight", and line 700 "delightfully" I would suggest removing these or replacing them with more academic phrases such as "indeed" or "as expected".

For the two parts we have refrained our emotion and rephrased them according to the suggestion as follows:

“As expected, the anti-FLAG M2 antibody was indeed found to be capable of recognizing HP-NAP as analyzed by ELISA (Fig. 2a) and Western blot (Fig. 2b).”

“We indeed found a correlation of 0.63 when examining the “wide type” + 5 mutants and 0.73 for the 5 mutants (Table S11), which could inspire a further examination of the usefulness of such a property for a larger set of isolated peptides in the future”

4. Fig. 6, extra borders in figures 6b and 6c.

Many thanks for the careful reading. After inspecting the whole content of main-text and supplementary material, we have removed the extra borders in following three figures: Figures 2, 5, 6b and 6c.

Fig. 5. The CD spectra from 190 to 260 nm for all the examined peptide in the study where CM15^{1,59} is a known helical AMP and Tat, a cell penetrating peptide (YGRKKRRQRRR) extracted from the transactivating transcriptional activator, is in a random coil structure⁵⁷.

Fig. 6. Helicity percentage as a function of helical propensity, peptide concentration|and average residue contact for the isolated peptide set (AMPs) and the TP-DB-stored peptides. (a, Up) The

5. line 619, "for each of"

We have corrected the phrase in that sentence as follows:

“According to Table 1, one can calculate the $\log(\text{HP}_{\text{NA}})$ value, helical propensity, for each of these 37 peptides.”

6. The figure resolution appears low, please improve.

In this revised manuscript, we have updated the Figures 1, 2, S4 and S5 with a higher resolution as follows:

a

Peptide _p :	ADEKKFWGKYLEVA	Peptide _z :	TAFEGGILKKGHHCSYTKH
Positions:	012345678901234	Positions:	0123456789012345678

b

Keys in Peptide _p	Start Positions in Peptide _p
A0D0E, D0E0K, ..., Y0E0V, E0V0A	0, 1, ..., 11, 12
A0D1K, D0E1K, ..., L0Y1V, Y0E1A	0, 1, ..., 10, 11
...	...
A0D4W, D0E4G, ..., G0K4V, K0Y4A	0, 1, ..., 7, 8
...	...
A4F0W, D4W0G, ..., G4E0V, K4V0A	0, 1, ..., 7, 8
...	...
A4F4L, D4W4Y, ..., K4K4V, K4Y4A	0, 1, ..., 3, 4

c

```
DB_INDEX = { A0D0E: [ [Peptidep, 0], [Peptidep+1, 17, 21, ...], ...],
             D0E0K: [ [Peptidep, 1], [Peptidep+9, 13, 25, ...], ...],
             ...,
             K4V0A: [ [Peptidep, 8], [Peptidep+78, 1, 64, ...], ...],
             ...,
             K4Y4A: [ [Peptidep, 4], [Peptidep+2, 23, 41, ...], ...],
             ... }
```

d

Sample patterns of interest	Keys in the DB	Examples of peptides that the patterns can match
ADE	A0D0E	ADEKKFWGKYLEVA
DEK	D0E0K	ADEKKFWGKYLEVA
K----VA	K4V0A	ADEKKFWGKYLEVA
K----Y----A	K4Y4A	ADEKKFWGKYLEVA

e

Simple Query: K 4 Y 4 A, which is equivalent to K4Y4A or K----Y----A
 Direct Fetch: K4Y4A from DB_INDEX in the RAM: →
 [[Peptide_p, 4], [Peptide_{p+2}, 23, 41, ...], ...],

f

[1] Complex Query: Process into a Combination of Simple Queries

Example of Complex Queries	Matching Patterns	Constituent Simple Queries
A 3 G 2 K 3 H 4 K	A 3 G 2 K 3 H 4 K	A 3 G 2 K K 3 H 4 K
A/Y 3 G 2 K 3 H 4 K	A 3 G 2 K 3 H 4 K	A 3 G 2 K K 3 H 4 K
	Y 3 G 2 K 3 H 4 K	Y 3 G 2 K K 3 H 4 K
A 2,3 G 2 K 3,4 H 4 K	A 2 G 2 K 3 H 4 K	A 2 G 2 K K 3 H 4 K
	A 2 G 2 K 4 H 4 K	A 2 G 2 K K 4 H 4 K
	A 3 G 2 K 3 H 4 K	A 3 G 2 K K 3 H 4 K
	A 3 G 2 K 4 H 4 K	A 3 G 2 K K 4 H 4 K

[2] Simple Queries' results are aggregated into the required results

7. Rephrase or perhaps remove line 705-707.

We have removed the statement from this section.

Reviewer #2 (Remarks to the Author):

The authors have addressed my prior comments.

Happy to learn that.

Reviewer #3 (Remarks to the Author):

Dear colleagues,

Your responses to the comments of all reviewers appear to address most of the concerns. I think this manuscript will be useful for helical therapeutic discoveries.

We thank for the Reviewer's appreciation.

--

Lee-Wei Yang, Professor,
 Director, Institute of Bioinformatics and Structural Biology,
 Director, PhD Program in Biomedical Artificial Intelligence,
 National Tsing Hua University,
 Honorary Professor, University of Liverpool, UK (Dual Degree PhD Program),
 Visiting Professor, University of Osaka, Japan (2018),
 Core member/recruitment committee, TIGP CBMB and Bioinfo Programs,
 Academia Sinica, Taiwan
 Group coordinator, Complex Systems and Mathematical Biology,
 National Center for Theoretical Sciences (Physics Division)
 101, Sec 2, Kuang-Fu Rd., Hsinchu, Taiwan, R.O.C.
 Voice: +886-3-574-2467, +886-3-571-5131 ext. 62452
 Fax: +886-3-571-5934
lwyang@life.nthu.edu.tw
dyn.life.nthu.edu.tw

Reviewers' Comments:

Reviewer #1:

Remarks to the Author:

The authors have adequately addressed most of my comments and the manuscript has been much improved. Helical 3D structures have also been added to the webserver with improved functionality and visualization. Below are a few additional comments regarding the new multiple regression model in the revision:

1. More rigorous mathematical description as well as implementation details of the multiple linear regression model, including formal statistical analysis/error estimation should be provided in the method.
2. line 626, the authors noted that that the peptide resolved at higher concentration (by NMR) has a higher % helical percentage but on line 635 and 640 the coefficient for the concentration is negative (-0.03).
3. Could the authors comment on the relative importance of helical propensity v. contact score v. concentration for determining helical percentage in the proposed models? Are the coefficients significant and interpretable?
4. Line 638-641, it is not clear to me how the error rate was determined (%). Was the error assessed by how well the model fitted the existing data or how well the model predicted new peptides not used for fitting?
5. The final model for the TB-DP database should be validated. If tested on the new data, the authors could show a scatter plot of predicted v. experimental values for helical percentage.
6. The linear model for the TP-DB was fitted using only 23 peptides but then used to estimate the helical percentages for the 1.7 million peptides Fig. S2(d). Although the training data seems well-sample, the author should comment if small sample size and sequence variability could limit model generalizability.
7. Line 700, was the new descriptors Max_sum4 fitted along with other parameters (e.g. helical propensity, contact number, concentration) in the linear model? The equation and coefficients should be provided.
8. The Max_sum4 descriptor with sliding windows appears to be promising. To tackle sequential data, have the authors also look into the applicability of deep learning frameworks such as CNN or RNN?

Minor comments:

1. Fig. S2D, missing x label, helical percentage (%), y label counts(N)
2. Fig. S2 line 52, NA (also elsewhere) should be subscripted.
3. line 306-307, "to our heartfelt delight", and line 700 "delightfully" I would suggest removing these or replacing them with more academic phrases such as "indeed" or "as expected".
4. Fig. 6, extra borders in figures 6b and 6c.
5. line 619, "for each of"
6. The figure resolution appears low, please improve.
7. Rephrase or perhaps remove line 705-707.

Reviewer #2:

Remarks to the Author:

The authors have addressed my prior comments.

Reviewer #3:

Remarks to the Author:

Dear colleagues,

Your responses to the comments of all reviewers appear to address most of the concerns. I think this manuscript will be useful for helical therapeutic discoveries.

Reviewer #1 (Remarks to the Author):

The authors have adequately addressed most of my comments and the manuscript has been much improved. Helical 3D structures have also been added to the webserver with improved functionality and visualization. Below are a few additional comments regarding the new multiple regression model in the revision:

We sincerely appreciate Reviewer 1's further suggestions that have help corroborate our arguments and refine the presentation.

1. More rigorous mathematical description as well as implementation details of the multiple linear regression model, including formal statistical analysis/error estimation should be provided in the method.

Thanks for pointing this out. We add the method associated with this in the SI.

Least-squares fitting to find helicity% as functions of helical propensity, concentration and tertiary contact

Let $\vec{x}_i = [1, x_1^i, x_2^i]$ where x_1 is the helical propensity and x_2 is either concentration or tertiary contact, where i is the index for individual peptides. The helicity percentage y_i can be expressed as a dot product of the independent variable \vec{x}_i and their parameters $\vec{\beta} = [\beta_0, \beta_1, \beta_2]$, where β_1 is the coefficient of x_1 and β_2 is the coefficient of x_2 . We can write

$$y_i \approx \sum_{j=0}^2 \beta_j \times x_j^i = \vec{\beta} \cdot \vec{x}_i \quad (\text{S1})$$

such that the experimentally observed or MD-determined helicity% on the left of the equal sign can be approximated by the dot product on the right. Here, $i=1$ to 37 or 41 for the AMP case and $i=1$ to 23 for isolated TP-DB peptides and therefore the n in the **Eq. S2** below is 37, 41 or 23.

In the least-squares fitting¹⁹, the optimum parameters $\vec{\beta}$ can be obtained from the minimized sum of mean squared loss $\sum_{i=1}^n (\vec{\beta} \cdot \vec{x}_i - y_i)^2$ such that

$$\vec{\beta} = \text{arg}_{\vec{\beta}} \min \sum_{i=1}^n (\vec{\beta} \cdot \vec{x}_i - y_i)^2 = (\mathbf{X}^T \mathbf{X})^{-1} \mathbf{X}^T \mathbf{Y} \quad (\text{S2})$$

where \mathbf{X} is a $n \times 3$ matrix constituted with the helical propensity and concentration (or tertiary contact) of n peptides and \mathbf{Y} is a $n \times 1$ matrix comprising n data points of experimentally observed or MD-determined helicity%. The matrix calculations were carried out using the programming language MATLAB.

The error estimate is further explained in Q4.

2. line 626, the authors noted that the peptide resolved at higher concentration (by NMR) has a higher % helical percentage but on line 635 and 640 the coefficient for the concentration is negative (-0.03).

We appreciate the careful reading. In our data set the average helicity% for the 37 NMR-determined short peptide structures is 0.687 and that for the 4 AMPs (namely CM15, W3_db5, W3_p1 and W3_p2) measured by CD in our study is 0.558, while the average concentration of the 37 peptides is 1.584 mM and that for our 4 AMPs is 0.06 mM. Collectively, we indeed see in average the NMR-solved structures of isolated peptides have a higher helicity% at a higher concentration (25 fold higher than that used in our CD experiments). However, as the Reviewer kindly noted that we do not see the concentration in mM range positively correlated with the helicity% (the coefficient for concentration is -0.03) when all the peptides used are structurally solved by NMR. We start to see that coefficient start to move toward the positive side (-0.009) while still being smaller than zero. Actually the coefficient becomes +0.043 if another three non-AMP peptides (namely, Tat, W3_n1 and W3_n2) we used in the CD experiment are included. Considering the unbalanced quantity between NMR-solved and CD-solved structures of the isolated peptides, we would like to keep the current regression results while noting that the equations are more suitable for isolated peptides of which the helicity% is examined by NMR. We added the following text with appreciating this limitation (before Figure 6).

“We also noted that the isolated peptide set does not exhibit expected dependency between helicity % and concentration, evidenced by the -0.03 coefficient for concentration. However, as only 4 CD-measured AMP data are included the correlation starts to grow toward the positive end (-0.009). In fact, if another three non-AMP peptides (tat, W3_n1 and W3_n2) are included, the pre-concentration coefficient becomes +0.043. Given the unbalanced data where our equations resulted from NMR-solved structures, these equations are more suitable to predict helicity% observed by NMR experiments.”

3. Could the authors comment on the relative importance of helical propensity v. contact score v. concentration for determining helical percentage in the proposed models? Are the coefficients significant and interpretable?

Thank the Reviewer's comment. We have addressed the concentration issue in the Q2 above. We see the helicity% for isolated helices derived from TP-DB could have a general trend. Along with that general trend, the sign and magnitude of coefficients for helical propensity and contact score are interpretable and agree with our hypothesis. We added a sentence in the Discussion (before Caveat).

“In the above regression models, the coefficients before helical propensity for the isolated peptide set and TP-DB peptide set are +0.067 and +0.076, respectively, agreeing with our expectation that observed helicity% grows with $\log(HP_{NA})$. In similar vein, the coefficients before average residue contact is -0.037, which also agrees with our earlier surmise that higher helical contact in peptides' original protein environment may lead to a slightly lower helicity% in their isolation.”

4. Line 638-641, it is not clear to me how the error rate was determined (%). Was the error assessed by how well the model fitted the existing data or how well the model predicted new peptides not used for fitting?

By "how well the model fitted the existing data". We now add the definition of r.m.s.d in the footnote of the new Table S12 and refer readers to the details in the footnote of Table S12, when we mentioned the 16.1% estimated error.

5. The final model for the TB-DP database should be validated. If tested on the new data, the authors could show a scatter plot of predicted v. experimental values for helical percentage.

We thank the good question. To address this, we did the same search as how we have collected the “isolated peptide set” except we required the date to be after Sept 1st, 2021 (before which we collected similar data to construct our regression model). We found only two peptides that are shorter than 30 amino acids containing helices (but not 100% helical) as well as containing neither disulfide bonds nor other materials (including nanoparticles) to stabilize their helices. The two isolated peptides are listed in Table S12 and they have an estimated error of 13.4%, which is slightly lower than the estimated error shown in Figure 6. We added the following sentences before Figure 6 –

“A month after we constructed this two-variable model, we found two extra NMR-determined isolated peptides that could also belong to the “isolated peptide set”, through which we could validate the model with an estimated error of 13.4%.”

6. The linear model for the TP-DB was fitted using only 23 peptides but then used to estimate the helical percentages for the 1.7 million peptides Fig. S2(d). Although the training data seems well-sampled, the author should comment if small sample size and sequence variability could limit model generalizability.

Yes, indeed. We have acknowledged the limitation as the Reviewer suggested in the section on isolated TP-DB peptides in Discussion by adding a sentence “**However, we should still note that the small sample size presented herein could limit the model generalizability.**”.

7. Line 700, were the new descriptors Max_sum4 fitted along with other parameters (e.g. helical propensity, contact number, concentration) in the linear model? The equation and coefficients should be provided.

We fitted it alone not with other parameters. The new equation has been added to the footnote of Table S11.

(Helicity% = 3.70 Max_sum4 - 2.82)

8. The Max_sum4 descriptor with sliding windows appears to be promising. To tackle sequential data, have the authors also look into the applicability of deep learning frameworks such as CNN or RNN?

We well appreciate the suggestion. The 4-gram feature can be of use to the suggested AI tests in similar studies. However, in our case, it can be somewhat challenging simply because the amount of labeled data (namely, helicity% for solvated short helical peptides in isolation) are too few (we had only 37 peptides in Table S10) for us to perform a good NN model.

Minor comments -

1. Fig. S2D, missing x label, helical percentage (%), y label counts(N)

Now both x and y labels are added in the Fig S2D as follows:

2. Fig. S2 line 52, NA (also elsewhere) should be subscripted.

Much thanks for the correction. After inspecting the whole content of main-text and supplementary material, two “NA” are revised to be subscripted in the caption of Fig. S2 and Table S11 as follows:

“(d) The estimated helicity % for TP-DB, excluding proline-contained peptides. The estimation is based on the relation $\text{Helicity Percentage} = 0.597 - 0.037 * \text{Avg. Res. Contact} + 0.076 * \log(\text{HP}_{\text{NA}})$, established by MD simulation results of 23 TP-DB peptides (see Fig 6).”

“Residues underlined are the location of folding cores, having the highest sum of $\log(\text{HP}_{\text{NA}})$ values (Max_sum4) among all the 4-residue windows in that peptide.”

3. line 306-307, "to our heartfelt delight", and line 700 "delightfully" I would suggest removing these or replacing them with more academic phrases such as "indeed" or "as expected".

For the two parts we have refrained our emotion and rephrased them according to the suggestion as follows:

“As expected, the anti-FLAG M2 antibody was indeed found to be capable of recognizing HP-NAP as analyzed by ELISA (Fig. 2a) and Western blot (Fig. 2b).”

“We indeed found a correlation of 0.63 when examining the “wide type” + 5 mutants and 0.73 for the 5 mutants (Table S11), which could inspire a further examination of the usefulness of such a property for a larger set of isolated peptides in the future”

4. Fig. 6, extra borders in figures 6b and 6c.

Many thanks for the careful reading. After inspecting the whole content of main-text and supplementary material, we have removed the extra borders in following three figures: Figures 2, 5, 6b and 6c.

Fig. 5. The CD spectra from 190 to 260 nm for all the examined peptide in the study where CM15^{1,59} is a known helical AMP and Tat, a cell penetrating peptide (YGRKKRRQRRR) extracted from the transactivating transcriptional activator, is in a random coil structure⁵⁷.

Fig. 6. Helicity percentage as a function of helical propensity, peptide concentration and average residue contact for the isolated peptide set (AMPs) and the TP-DB-stored peptides. (a, Up) The

5. line 619, "for each of"

We have corrected the phrase in that sentence as follows:

“According to Table 1, one can calculate the $\log(\text{HP}_{\text{NA}})$ value, helical propensity, for each of these 37 peptides.”

6. The figure resolution appears low, please improve.

In this revised manuscript, we have updated the Figures 1, 2, S4 and S5 with a higher resolution as follows:

a

Peptide _p :	ADEKKFWGKYLEVA	Peptide _z :	TAFEGGILKKGHHCSYTKH
Positions:	012345678901234	Positions:	0123456789012345678

b

Keys in Peptide _p		Start Positions in Peptide _p
A0D0E, D0E0K, ..., Y0E0V, E0V0A		0, 1, ..., 11, 12
A0D1K, D0E1K, ..., L0Y1V, Y0E1A		0, 1, ..., 10, 11
...		...
A0D4W, D0E4G, ..., G0K4V, K0Y4A		0, 1, ..., 7, 8
...		...
A4F0W, D4W0G, ..., G4E0V, K4V0A		0, 1, ..., 7, 8
...		...
A4F4L, D4W4Y, ..., K4K4V, K4Y4A		0, 1, ..., 3, 4

c

```
DBINDEX = { A0D0E: [ [Peptidep, 0], [Peptidep+1, 17, 21, ...], ...],
             D0E0K: [ [Peptidep, 1], [Peptidep+9, 13, 25, ...], ...],
             ...
             K4V0A: [ [Peptidep, 8], [Peptidep+78, 1, 64, ...], ...],
             ...
             K4Y4A: [ [Peptidep, 4], [Peptidep+2, 23, 41, ...], ...],
             ... }
```

d

Sample patterns of interest	Keys in the DB	Examples of peptides that the patterns can match
ADE	A0D0E	ADEKKFWGKYLEVA
DEK	D0E0K	ADEKKFWGKYLEVA
K----VA	K4V0A	ADEKKFWGKYLEVA
K----Y----A	K4Y4A	ADEKFWGKYLEVA

e

Simple Query: K 4 Y 4 A, which is equivalent to K4Y4A or K----Y----A
 Direct Fetch: K4Y4A from DB_{INDEX} in the RAM: →
 [[Peptide_p, 4], [Peptide_{p+2}, 23, 41, ...], ...],

f

[1] Complex Query: Process into a Combination of Simple Queries

Example of Complex Queries	Matching Patterns	Constituent Simple Queries
A 3 G 2 K 3 H 4 K	A 3 G 2 K 3 H 4 K	A 3 G 2 K K 3 H 4 K
A/Y 3 G 2 K 3 H 4 K	A 3 G 2 K 3 H 4 K	A 3 G 2 K K 3 H 4 K
	Y 3 G 2 K 3 H 4 K	Y 3 G 2 K K 3 H 4 K
A 2,3 G 2 K 3,4 H 4 K	A 2 G 2 K 3 H 4 K	A 2 G 2 K K 3 H 4 K
	A 2 G 2 K 4 H 4 K	A 2 G 2 K K 4 H 4 K
	A 3 G 2 K 3 H 4 K	A 3 G 2 K K 3 H 4 K
	A 3 G 2 K 4 H 4 K	A 3 G 2 K K 4 H 4 K

[2] Simple Queries' results are aggregated into the required results

7. Rephrase or perhaps remove line 705-707.

We have removed the statement from this section.

Reviewer #2 (Remarks to the Author):

The authors have addressed my prior comments.

Happy to learn that.

Reviewer #3 (Remarks to the Author):

Dear colleagues,

Your responses to the comments of all reviewers appear to address most of the concerns. I think this manuscript will be useful for helical therapeutic discoveries.

We thank for the Reviewer's appreciation.

Reviewers' Comments:

Reviewer #1:

Remarks to the Author:

The authors have addressed all of my comments and have no further comments at this point.

Reviewer #2:

None

Reviewer #1 (Remarks to the Author):

The authors have addressed all of my comments and have no further comments at this point.

We thank for the Reviewer's appreciation.